# CEP41-mediated ciliary tubulin glutamylation drives angiogenesis through AURKA-dependent deciliation

Soo Mi Ki[1,†], Ji Hyun Kim[1,†], So Yeon Won[1,†], Shin Ji Oh[1], In Young Lee[2], Young-Ki Bae[3], Ki Wha Chung[4], Byung-Ok Choi[5], Boyoun Park[6], Eui-Ju Choi[2] & Ji Eun Lee[1,7,*] (iD)

## Abstract

The endothelial cilium is a microtubule-based organelle responsible for blood flow-induced mechanosensation and signal transduction during angiogenesis. The precise function and mechanisms by which ciliary mechanosensation occurs, however, are poorly understood. Although posttranslational modifications (PTMs) of cytoplasmic tubulin are known to be important in angiogenesis, the specific roles of ciliary tubulin PTMs play remain unclear. Here, we report that loss of centrosomal protein 41 (CEP41) results in vascular impairment in human cell lines and zebrafish, implying a previously unknown pro-angiogenic role for CEP41. We show that proper control of tubulin glutamylation by CEP41 is necessary for cilia disassembly and that is involved in endothelial cell (EC) dynamics such as migration and tubulogenesis. We show that in ECs responding to shear stress or hypoxia, CEP41 activates Aurora kinase A (AURKA) and upregulates expression of VEGFA and VEGFR2 through ciliary tubulin glutamylation, as well as leads to the deciliation. We further show that in hypoxia-induced angiogenesis, CEP41 is responsible for the activation of HIF1α to trigger the AURKA-VEGF pathway. Overall, our results suggest the CEP41-HIF1α-AURKA-VEGF axis as a key molecular mechanism of angiogenesis and demonstrate how important ciliary tubulin glutamylation is in mechanosense-responded EC dynamics.

**Keywords** angiogenesis; AURKA; CEP41; primary cilia; tubulin glutamylation
**Subject Categories** Cell Adhesion, Polarity & Cytoskeleton; Vascular Biology & Angiogenesis

## Introduction

Angiogenesis is the process by which new vessels grow from existing vasculature. It is orchestrated by extracellular cues that trigger dynamic vascular EC behaviors, such as invasion, migration, proliferation, differentiation, and morphogenesis [1]. Angiogenesis is vital for supplying oxygen and nutrients to body tissues [1] and is reportedly involved in the pathogenesis of several disorders [2]. Upon activation by angiogenic signals, ECs in pre-existing vessels sprout new branches with directional specificity to extend the vascular network. During angiogenic branching, migrating ECs in a sprout are morphologically polarized to facilitate directional migration. This process is tightly regulated by microtubules [3]. Indeed, the proper control of EC microtubule growth is essential not only for producing new sprouts but also for supporting the existing branches [4].

Microtubules are filamentous polymer components of the cytoskeleton, which regulates cell shape. These cylindrical polymers, composed of repeating α/ß tubulin heterodimer subunits, are responsible for many cellular phenomena, including cell division, the organization of intracellular structures, and ciliary function. Because microtubules are polar filaments, having their minus-ends anchored at the centrosome and their plus-ends extended to the cell periphery, they function in a directional manner. During angiogenesis, the cell polarization and migration events that occur in its earliest branching stages require the structural and functional framework provided by the EC microtubule network [5–7].

The centrosome, which is composed of microtubule-based centrioles, is essential for coordinated signaling during cell polarization and migration. Centrosome-controlled reorientation of vascular ECs toward the intended migratory direction is crucial for early angiogenesis [8], especially during vascular proliferation and sprouting angiogenesis [9,10]. Although our understanding of the role the

1  Department of Health Sciences and Technology, SAIHST, Sungkyunkwan University, Seoul, South Korea
2  Laboratory of Cell Death and Human Diseases, Department of Life Sciences, Korea University, Seoul, South Korea
3  Comparative Biomedicine Research & Tumor Microenvironment Research Branch, Research Institute, National Cancer Center, Goyang, Korea
4  Department of Biological Sciences, Kongju National University, Kongju, South Korea
5  Department of Neurology, Sungkyunkwan University School of Medicine, Seoul, South Korea
6  Department of Systems Biology, College of Life Science and Biotechnology, Yonsei University, Seoul, South Korea
7  Samsung Biomedical Research Institute, Samsung Medical Center, Seoul, South Korea
   *Corresponding author. Tel: +82 2 3410 6129; E-mail: jieun.lee@skku.edu
   †These authors contributed equally to this work

centrosome itself plays during angiogenesis is becoming more clear, the roles individual centrosomal proteins play in the process have received much less attention.

The cilium, which is assembled on a mother centriole, protrudes from the surface of most eukaryotic cells where it is involved in mechanosensing, extracellular signal transduction, cell division, and migration [11]. *In vitro* studies have suggested vascular ECs sense and transduce biomechanical stimuli (e.g., blood flow-induced shear stress) through their cilia [12–14]. Recent *in vivo* studies in the zebrafish model system have demonstrated that endothelial cilia are essential for both the transduction of blood flow-dependent mechanosignals and the Hedgehog signal that are required for the development of the vascular network [15,16]. The molecular mechanisms regulating cilia-dependent mechanotransduction and function, however, remain poorly understood.

The ciliary axoneme, a highly dynamic microtubule-based structure, undergoes several types of PTMs including (de)acetylation, (de)tyrosination, (de)glycylation, and (de)glutamylation. Tubulin PTMs have been implicated in diverse microtubule-related functions and disease states [17–19]. With respect to ciliary dynamics and function, acetylation is crucial for cilia assembly [20], and glutamylation seems to be critical for cilia stability and motility [19,21]. Although a role for the (de)acetylation of cytoplasmic tubulin has recently been discovered in angiogenesis [22,23], it is unclear whether other PTMs of ciliary tubulin also regulate angiogenesis.

CEP41 is a ciliary protein associated with the ciliopathy referred to as Joubert syndrome [21]. CEP41 is essential for tubulin glutamylation in the cilia but not in the cytoplasm, and this glutamylation is essential for the maintenance of ciliary structure and motility in zebrafish [21]. Here, we sought to determine the EC-specific roles of CEP41 and clarify the importance of PTMs of ciliary tubulin in angiogenesis. We show that *CEP41* depletion inhibits angiogenesis and reduces glutamylation of the tubulin in EC cilia both *in vitro* and *in vivo*. We provide evidence that CEP41-mediated tubulin glutamylation leads to the disassembly of endothelial cilia via AURKA-activated mechanotransduction. We further demonstrate that CEP41 binds HIF1α and their physical interaction is required for the activation of AURKA and its resulting regulation of EC dynamics during angiogenesis. Our results clarify the role of tubulin glutamylation in EC cilia plays in angiogenesis. Our results also reveal the mechanism by which EC deciliation induced by mechanosensation facilitates the expression of pro-angiogenic regulators.

# Results

## Angiogenesis requires CEP41 because of its regulation of EC dynamics

To determine whether CEP41 plays a role in angiogenesis, we examined the effect of *CEP41* depletion on EC behavior using validated *CEP41* siRNAs (Appendix Fig S1) in human umbilical vein endothelial cells (HUVECs). We first assessed the role of CEP41 on cell migration, a process critical for EC remodeling, using an *in vitro* wound healing assay. We found that 12 h after wound induction by scratching, control siRNA-transfected cells show nearly 80% wound closure, whereas both types of *CEP41*-depleted cells show levels of wound closure roughly 50% of the controls (Fig 1A and B). We then performed a transwell cell invasion assay and found significantly less invasion in *CEP41*-deficient HUVECs than controls (Fig 1C and D). Next, to further clarify the effect of *CEP41* depletion-induced cellular defects on angiogenesis, we performed an *in vitro* tube formation assay. After observing HUVECs seeded onto Matrigel-coated plates for 18 h, we found that while control cells form tubular networks of interconnected branches, this process is dramatically hampered in *CEP41*-depleted cells (Fig 1E). After further analysis of the number of tube nodes and cumulative tube length, we concluded that *CEP41* deficiency significantly attenuates tubulogenesis in ECs (Fig 1F and G). This suggests CEP41 is essential for vascular EC dynamics including migration, invasion, and tubulogenesis.

Next, to identify the EC-specific role of CEP41 in angiogenesis *in vivo*, we performed a *loss-of-function* study using zebrafish model in which the vasculature consists of an EC network until 72 h post-fertilization (hpf) [24]. To monitor blood vessel development, we used zebrafish whose vascular ECs are labeled with enhanced GFP. These *Tg(kdrl:eGFP)*^*s843* zebrafish [25] are referred to hereafter as *Tg(kdrl:eGFP)*. We generated *cep41*-knockdown and *cep41*-knockout zebrafish using verified AUG morpholinos (MOs) [21], splice-blocking MOs (Appendix Fig S2A–E), and the CRISPR/Cas9 gene-editing system (Appendix Fig S2F–I). The intersegmental vessels (ISVs) in zebrafish are formed from 24 hpf when the ECs migrate dorsally from the dorsal aorta (DA) (Appendix Fig S2J). As in human cell lines, both *cep41* mutants and morphants show pronounced vascular defects, such as thin, short, fused, or missing ISVs (Fig 2A–C), as well as ruptured dorsal longitudinal anastomotic vessels (DLAVs) at 40 hpf (Fig 2D and E). We further investigated the role of CEP41 in sprouting angiogenesis in the caudal vein

---

**Figure 1. *CEP41* depletion restricts endothelial cell behavior.**

A, B  HUVECs transfected with control or *CEP41* siRNAs were scratched (0 h) to induce wounding and then incubated for 12 h to allow wound closure. The wound margins were observed every 4 h in the *CEP41*#1 and #2 siRNA-transfected cells and compared to those of control cells. Representative images of cells subjected to the wound closure assay in (A). Scale bars, 600 μm. Quantification of the extent of wound closure in (B) presented graphically by measuring the distance between the dotted lines at each time point. Data are shown as mean ± SD of three independent experiments (*n* ≥ 3 scratches per experimental condition). Statistical significance was assessed using the two-way ANOVA followed by Tukey's *post hoc* test (***$P < 0.001$).

C, D  The siRNA-transfected HUVECs were plated inside a transwell chamber and incubated with serum for 18 h. The cells that invaded were observed after staining with crystal violet (CV) solution. Scale bars, 600 μm. The numbers of cells that invaded in each field of view were counted with the ImageJ software in (D). The data indicate the results of three independent experiments with ≥ 3 invasions per condition (mean ± SD). ***$P < 0.001$ (one-way ANOVA with Tukey's *post hoc* test).

E–G  Tubulogenesis of control and *CEP41*-knockdown cells for 18 h was compared via an *in vitro* angiogenesis assay. Scale bars, 600 μm. Quantification of tube node numbers in (F) and tube length in (G) from each field of view using the ImageJ angiogenesis analyzer at the indicated time points. The graph compares the relative length of control cells and *CEP41*-knockdown cells. Data are shown as mean ± SD of five independent experiments with ≥ 5 tubulogenesis regions per condition. Statistical significance was assessed with the two-way ANOVA followed by Tukey's *post hoc* test (**$P < 0.01$, ***$P < 0.001$).

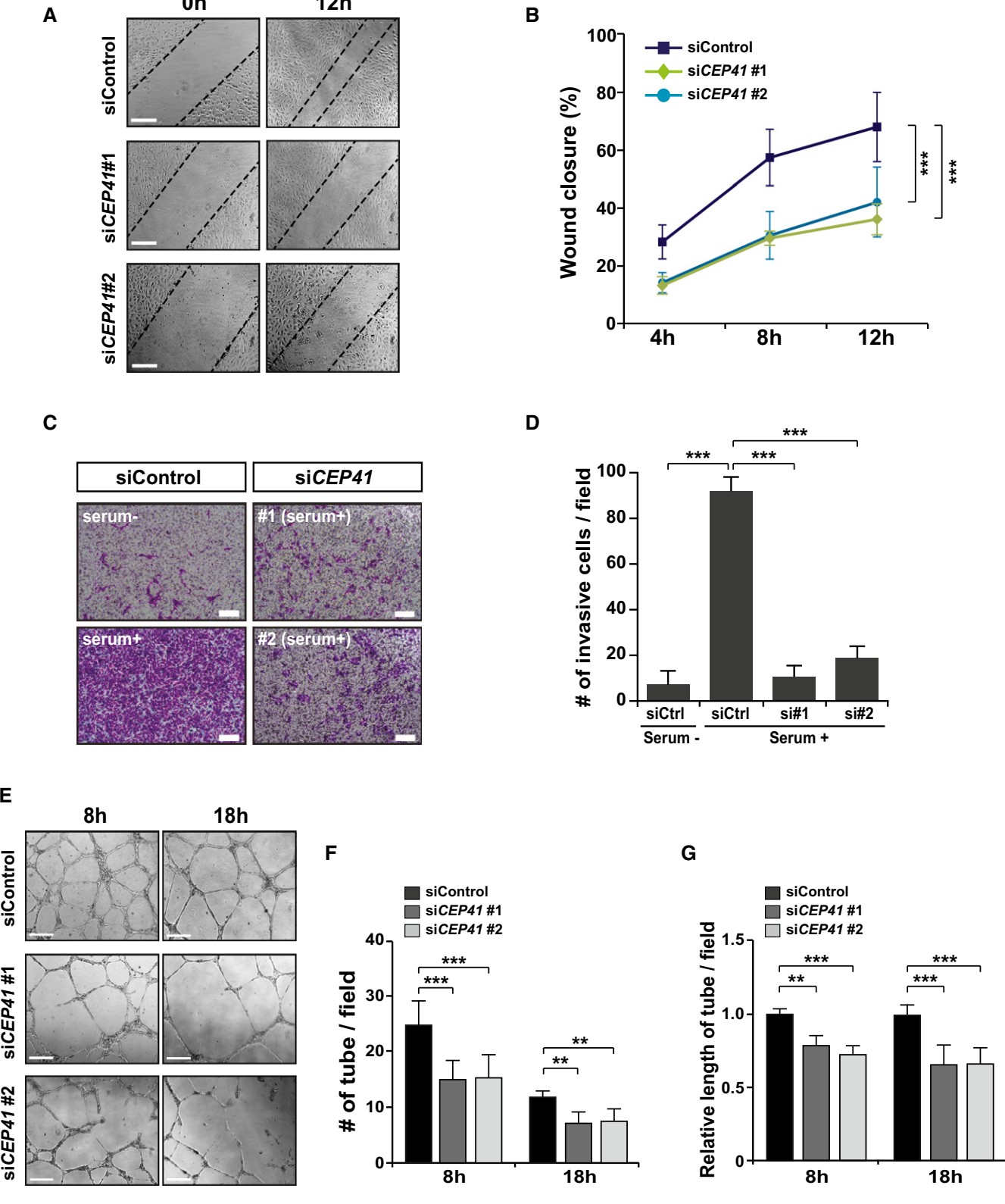

**Figure 1.**

plexus (Appendix Fig S2J), the venous ECs of which undergo highly dynamic remodeling to ultimately form a functional caudal vein (CV). We found the axial vessels in the caudal region of control embryos at 35 hpf show a network with spaces between the capillaries, whereas *cep41* mutant and morphant embryos fail to form this network by fusing their various capillaries (Fig 2F). In a more

detailed analysis of *cep41*-depleted zebrafish, we found reduced venous sprouts and vascular loops, implicating defects in EC remodeling in the disruption of CV formation (Fig 2F and G). In addition, we found that the aberrant vasculature of *cep41* morphants cannot be attributed to a general effect of the MOs on blood circulation because we did not observe any significant change in heart beating or blood flow (Appendix Fig S2K–M and Movie EV1–EV4). Together, our data suggest CEP41 plays a fundamental role in angiogenesis in humans and zebrafish through its effects on the remodeling of vascular ECs.

**Glutamylation of EC cilia is essential for angiogenesis**

Previously, we demonstrated that CEP41 is dispensable for initial cilia assembly but required for ciliary tubulin glutamylation in epithelial cells [21]. Thus, we investigated the cilia-related role of CEP41 in vascular ECs. We transfected HUVECs with control or *CEP41*-specific siRNAs and immunostained them with antibodies specific to ARL13b (as a cilia marker), γ-Tub (as a centrosomal marker), Ac-Tub (as an acetylated-tubulin marker), and GT335 (as a glutamylated-tubulin marker). We found that both control and *CEP41*-depleted cells are similarly positive for ARL13b and γ-Tub, suggesting CEP41 is not required for the initial formation of the centrosome or the cilium (Fig EV1A–D). Neither did we observe any difference in Ac-Tub staining between control and *CEP41*-depleted cells (Fig 3A and B). However, we found *CEP41*-depleted cells show a striking reduction in GT335-positive cilia (Fig 3A and B). Next, to determine this glutamylation-specific function for CEP41 in endothelial cilia *in vivo*, we immunostained *cep41*-knockdown and *cep41*-knockout *Tg(kdrl:eGFP)* zebrafish with antibodies specific to ARL13b, Ac-Tub, and GT335 at 28 hpf, when the endothelial cilia are visible in the caudal vein plexus. Consistent with our results in human ECs, *cep41*-ablated zebrafish showed a drastic reduction in cilia with glutamylated tubulin (Fig 3C and D) without any associated defects in cilia formation or tubulin acetylation (Figs 3C and D, and EV1E and F). Together, these results indicate CEP41 is important for the glutamylation tubulin in EC cilia.

According to recent studies, a deficit of cytoplasmic carboxypeptidase 5 (CCP5), a deglutamylase that acts on ciliary tubulin, restores the hypoglutamylation caused by a tubulin glutamylase deficiency in the ciliary axoneme [26,27]. Thus, to clarify the impact of ciliary tubulin glutamylation on angiogenesis, we measured the effects of *CCP5* depletion on *CEP41*-depleted EC dynamics. After transfecting HUVECs with validated *CCP5*-specific siRNAs (Appendix Fig S3A), we performed wound healing and tube formation assays. We found *CCP5* depletion attenuates cell migration and tubulogenesis, phenotypes similar to the angiogenic defects of *CEP41*-silenced cells (Fig 4A–E). Furthermore, we found HUVECs co-transfected with *CEP41*- and *CCP5*-specific siRNAs show a marked rescue of the defective cell migration and tubulogenesis phenotypes resulting from depletion of *CEP41* or *CCP5* alone (Fig 4A–E). Next, we injected *cep41*-knockdown or *cep41*-knockout *Tg(kdrl:eGFP)* zebrafish with verified *ccp5* MOs [26] shown to affect zebrafish angiogenesis (Fig EV2). We found *ccp5* depletion rescues the impairments of ISVs and DLAVs caused by *cep41* deficiency at 40 hpf (Fig 4F and G). These results implicate the glutamylation of ciliary tubulin as an important regulator of EC dynamics in the control of angiogenesis. Finally, to further demonstrate the effect of *CCP5* silencing on the cilia with hypoglutamylated tubulin induced by *CEP41* depletion, we immunostained the co-transfected cells with both ARL13b-specific and GT335-specific antibodies. As we saw in *CEP41*-depleted cells, *CCP5* silencing, either alone or in addition to *CEP41* silencing, does not affect ciliation (ARL13b-labeled cells) (Fig 4H). Remarkably, *CCP5* silencing enhances the level of ciliary tubulin glutamylation and restores the reduced tubulin glutamylation of *CEP41*-depleted cilia (Fig 4I). Furthermore, we found that although control HUVECs produce ARL13b-positive cilia ~6 μm in length and GT335-positive cilia ~5 μm in length, *CEP41*-knockdown cells produce shorter cilia in both types (Fig 4J). In contrast, *CCP5* depletion led to the production of longer ARL13b- and GT335-positive cilia than control and, furthermore, restored the shortened cilia of *CEP41*-knockdown cells (Fig 4J). Together, these data suggest that the regulation of CEP41 and CCP5 is required for proper glutamylation of tubulin in cilia, which is necessary for precise control of ciliary length and in turn involved in control of angiogenesis.

**CEP41 facilitates deciliation by its role in mechanotransduction via endothelial cilia**

Given that cilia are required for mechanotransduction during EC remodeling [12,28], we next asked whether CEP41 regulates mechanosignal transduction in ECs. After transfecting HUVECs with control or *CEP41* siRNAs, we exposed them to laminar shear stress using a verified fluid flow model [29] (Appendix Fig S4A). We found the control cells show elongated and aligned morphologies after

**Figure 2. Ablation of *cep41* impairs vascular development in zebrafish.**

A   Blood vessels were observed in MOs (control, *cep41* AUG (2.5 ng), or *cep41* SB (2 ng))-injected or *cep41*-mutated *Tg(kdrl:eGFP)* zebrafish at 40 hours post-fertilization (hpf) by fluorescent microscopy. Asterisks and arrowheads indicate impaired ISVs and DLAVs, respectively. The representative images for analysis of ISV lumen diameter are indicated by dotted rectangles. A, anterior; P, posterior; DLAV, dorsal longitudinal anastomotic vessel; ISV, intersegmental vessel. Scale bars, 100 μm.

B–E   Quantification of ISV lumen diameter in (B), the numbers of defective ISVs in (C), the numbers of embryos with aberrant DLAVs in (D), and the numbers of ruptured DLAVs in (E) from data observed in equivalent fields of view (within eight somites). The severity of blood vessel defects in *cep41*-deficient zebrafish: B (narrowed ISVs) < C (shorten, fused, and missing ISVs) < D and E (ruptured DLAVs). Data are shown as mean ± SD of three independent experiments with ≥ 20 embryos per condition. Statistical significance was determined using the one-way ANOVA followed by Dunnett's *post hoc* test (B) and the Kruskal–Wallis test by Dunn's *post hoc* test (C, E) (**$P < 0.01$, ***$P < 0.001$).

F, G   CV protrusions (arrowheads) and vascular loops (asterisks) were observed in zebrafish at 35 hpf. The images within the dotted rectangles are magnified in the right panels. A, anterior; P, posterior; CA, caudal artery; CV, caudal vein. Scale bars, 40 μm. Quantification of the length of the CV protrusions in (G) presented graphically by measuring them in equivalent fields of view (within five somites). Data are median of four independent experiments ($n \geq 5$ embryos per experimental condition). Statistical significance was determined using the Kruskal–Wallis test followed by Dunn's *post hoc* test (***$P < 0.001$, ns: non-significant).

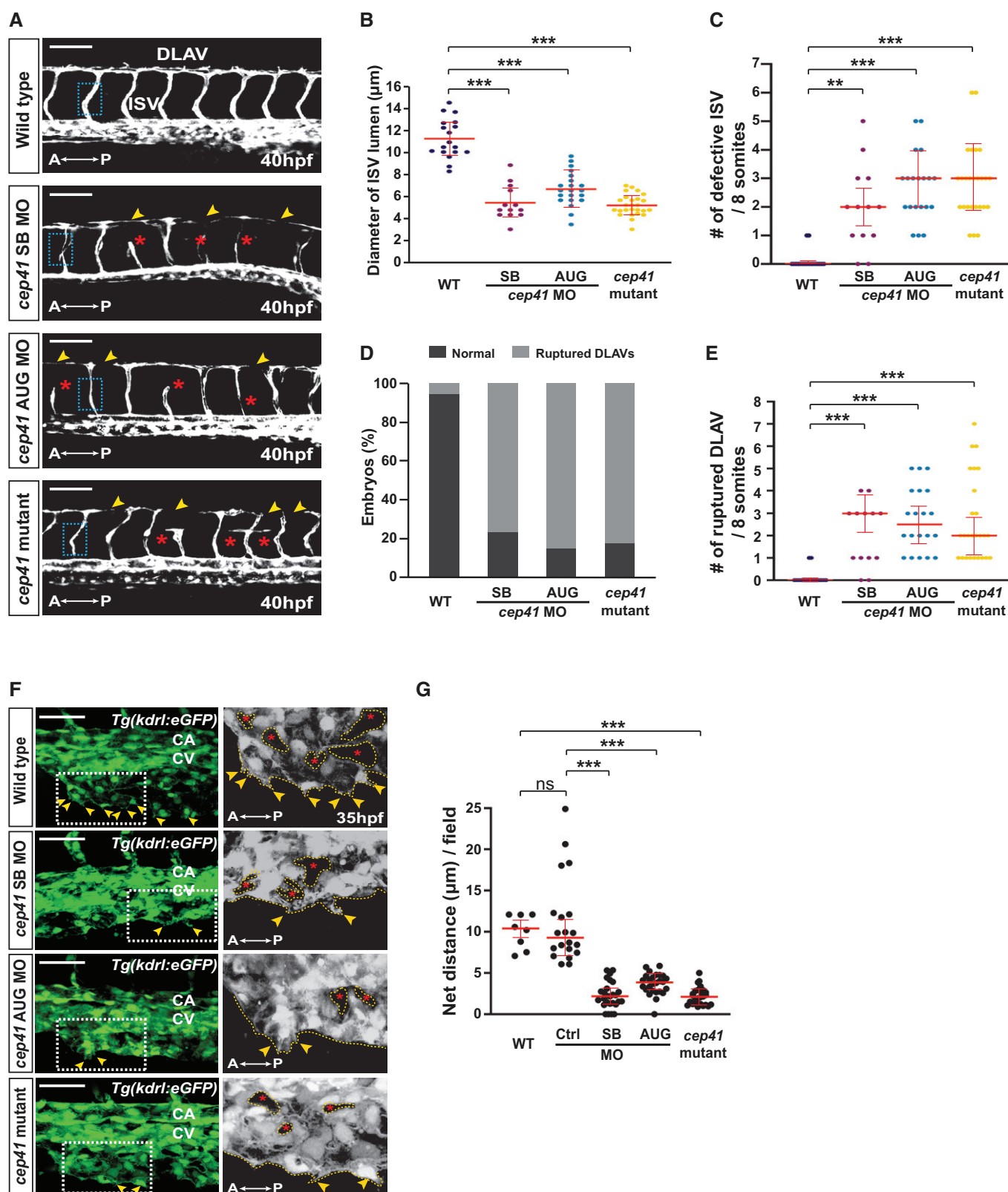

**Figure 2.**

exposure to shear stress (Appendix Fig S4C). In contrast, *CEP41*-deficient cells under shear stress retain the cobblestone appearance that control cells show under static conditions (Appendix Fig S4B and D), indicating defective mechanosensation. Next, to determine whether the altered mechanosignal transduction affects angiogenesis, we examined the effect of *CEP41* depletion on EC migration and

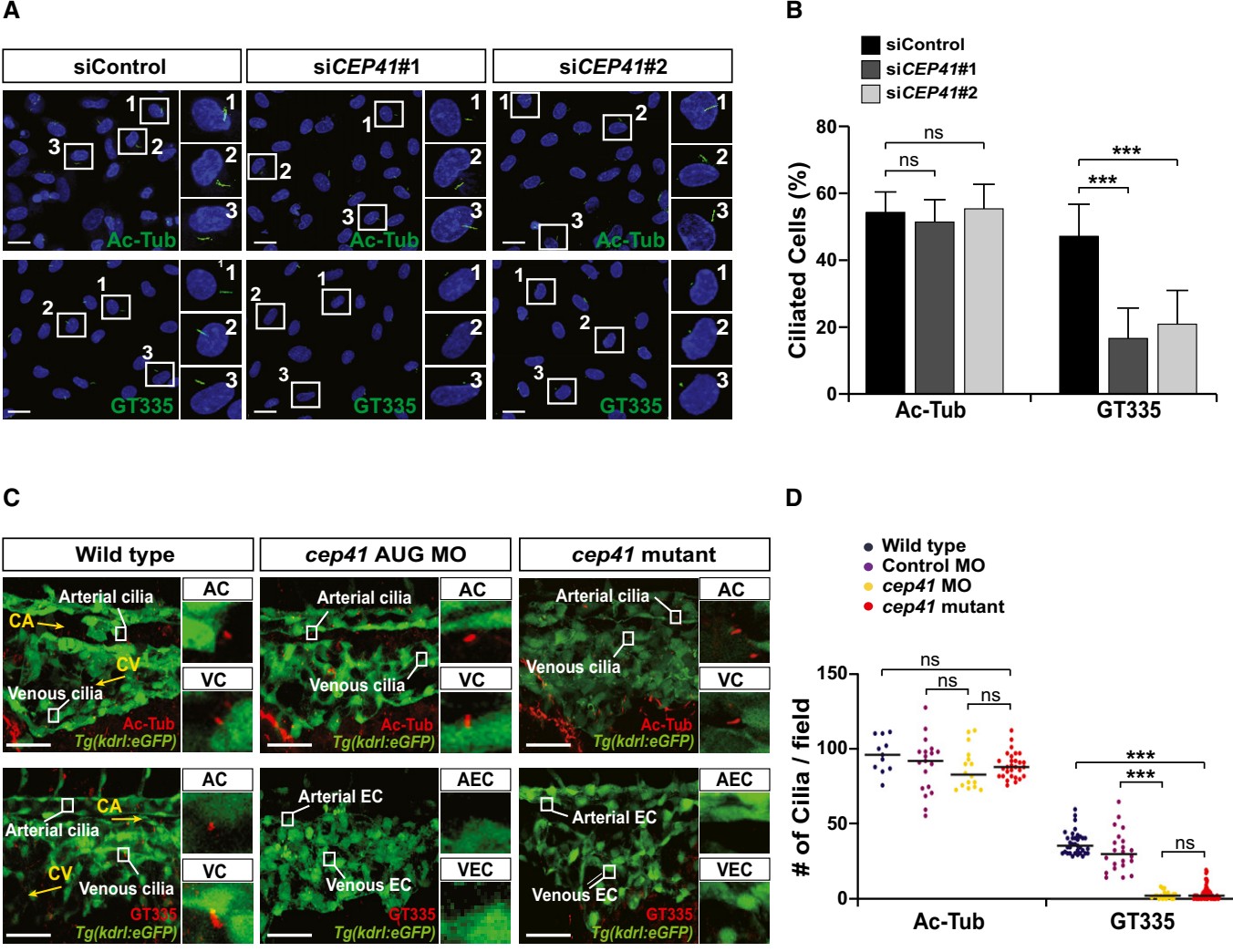

**Figure 3. CEP41 facilitates tubulin glutamylation in endothelial cilia.**

A, B  Control or *CEP41* siRNA-transfected HUVECs were immunostained with Ac-Tub- or GT335-specific antibodies in (A). The rectangles indicate the representative cells from each immunostaining experiment presented as magnified images in the right panels. Scale bars, 20 μm. Quantification of the ciliated cell numbers in the images in (A) from the results of three independent experiments with ≥ 200 cells per condition (mean ± SD) in (B). ***$P < 0.001$, ns: non-significant (one-way ANOVA with Dunnett's *post hoc* test).

C, D  Wild-type and *cep41*-deficient *Tg(kdrl:eGFP)* zebrafish embryos were fixed for immunostaining with Ac-Tub- or GT335-specific antibodies at 28 hpf in (C). The rectangles indicate the arterial cilia (AC) and venous cilia (VC) in zebrafish endothelial cells (EC). Magnified representative images are displayed in the right panels. Scale bars, 40 μm. Quantification of the labeled cilia observed in equivalent fields of view is presented graphically in (D). Data are shown as median of three independent experiments ($n ≥ 20$ embryos per condition). Statistical significance was determined using the Kruskal–Wallis test followed by Dunn's *post hoc* test (***$P < 0.001$, ns: non-significant).

tubulogenesis under shear stress conditions. We found control cell migration is significantly increased under laminar shear stress (≥ 15 dynes/cm$^2$), as previously reported [30,31], whereas *CEP41*-depleted cell migration is unaffected (Fig 5A and B). In addition, while control cells enhance tube formation by shear stress, *CEP41*-silenced cells show no change in tube formation under similar stress (Fig 5C–E). These data indicate a role for CEP41 in EC mechanotransduction and its influence on angiogenesis.

Endothelial cilia sense the onset of blood flow-induced shear stress, but it is unclear whether they are persistently required for angiogenesis. Previously, several *in vivo* studies reported that the presence of endothelial cilia depends on various physiological contexts and developmental stages in which diverse types of shear stress are produced [15,32,33]. Of note, high laminar shear stress induces microtubule depolymerization, which is critical for angiogenic EC remodeling [34,35] and cilia disassembly *in vitro* [36] and *in vivo* [33]. Consistent with these reports, we found HUVECs exposed to low shear stress (3.96 dynes/cm$^2$) show similar numbers of cilia to those of static control cells, but cells exposed to high shear stress (19 dynes/cm$^2$) reduce the numbers of cilia (Fig EV3A and B). Moreover, we found that the overall length of cilia retained in high shear stress-responded cells is shortened, whereas the length

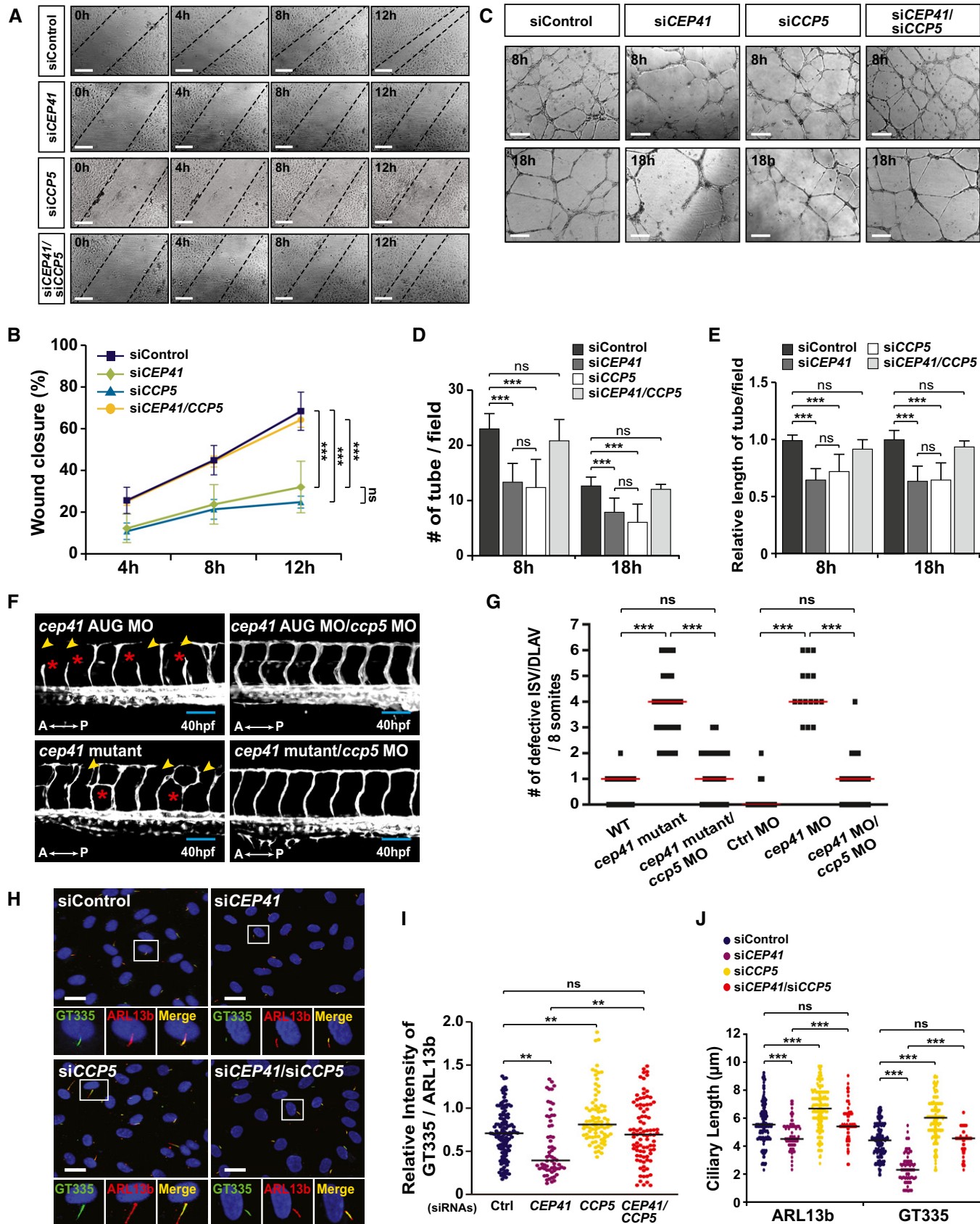

Figure 4.

**Figure 4.  Depletion of *CCP5* rescues the angiogenesis defects caused by *CEP41* deficiency.**

A, B    HUVECs transfected with *CCP5* and/or *CEP41* siRNAs were scratched and incubated to evaluate wound closure. The extent of wound closure in co-transfected cells with *CCP5* and *CEP41* siRNAs was compared with that in control or *CEP41* siRNA-transfected cells at the indicated time points (A). Scale bars, 600 μm. Quantification of the extent of wound closure in (A) is presented graphically by measuring the distance between the dotted lines at each time point in (B). Data are shown as mean ± SD of three independent experiments ($n \geq 3$ scratches per experimental condition). Statistical significance was determined using the two-way ANOVA followed by Tukey's *post hoc* test (\*\*\**P* < 0.001, ns: non-significant).

C–E    The indicated siRNA-transfected HUVECs were subjected to *in vitro* angiogenesis assays. Tubulogenesis of HUVECs depleted for *CEP41* and *CCP5* was compared to that of control, *CEP41* single-knockdown, or *CCP5* single-knockdown cells (C). Scale bars, 600 μm. Quantification of tube node number in (D) and tube length in (E) from data examined within equivalent fields of view at the indicated time points using the ImageJ angiogenesis analyzer. Data are shown as mean ± SD of five independent experiments with $\geq 5$ tubulogenesis regions per condition. Statistical significance was determined using the one-way ANOVA followed by Tukey's *post hoc* test (\*\*\**P* < 0.001, ns: non-significant).

F, G    The blood vessels in the trunks of *cep41*-mutant/morphant zebrafish were compared to those of zebrafish co-injected with *ccp5* MOs (2 ng) at 40 hpf in (F). Asterisks and arrowheads indicate impaired ISVs and DLAVs, respectively. A, anterior; P, posterior. Scale bars, 100 μm. Quantification of the number of defective ISVs and DLAVs from data observed in equivalent fields of view (within eight somites). Data are median of three independent experiments with $\geq 20$ embryos per condition. Statistical significance was determined using the Kruskal–Wallis test followed by Dunn's *post hoc* test (\*\*\**P* < 0.001).

H–J    HUVECs co-transfected with *CEP41* and *CCP5* siRNAs were immunostained with ARL13b- and GT335-specific antibodies. The staining results were compared to those of control, single *CEP41*, or *CCP5* siRNA-transfected cells. Representative images indicated with rectangles appear in magnified images in the lower panels. Scale bars, 20 μm. Quantification of relative intensity of GT335/ARL13b signals in the cilia (I) and ciliary length (J) in images (H) is the result of three independent experiments with $\geq 200$ cells per condition (median). \*\**P* < 0.01, \*\*\**P* < 0.001, ns: non-significant (Kruskal–Wallis test with Dunn's *post hoc* test).

of glutamylated cilia is not affected (Fig EV3C and D). Notably, high shear stress-exposed cilia show stronger glutamylation intensity than both controls and low shear stress-exposed cilia (Fig EV3E). We further found that while control cell cilia are largely resorbed, *CEP41* depletion attenuates the cilia disassembly of HUVECs following exposure to high shear stress (Fig 5F and G). These results, therefore, imply the glutamylation of ciliary tubulin influences EC deciliation under high shear stress conditions.

To more clarify the role of ciliary tubulin glutamylation plays in deciliation, we examined the effect of *CEP41* and *CCP5* depletion on cilia disassembly in human telomerase reverse transcriptase–retinal pigment epithelial (hTERT-RPE1) cells. These cells modulate the assembly/disassembly of their cilia according to serum conditions. After subjecting control RPE1 cells and *CEP41* or *CCP5* siRNA-transfected RPE1 cells to 48 h of serum starvation to induce cilia assembly, we applied a serum retrieval protocol for 18 h to induce cilia disassembly. By staining the cells with ARL13b-specific antibodies, we observed disassembly of ~50% of the cilia in control cells, but observed retention of most cilia in both *CEP41*- and *CCP5*-depleted cells (Fig 5I and J). Remarkably, co-transfection with *CEP41*- and

*CCP5*-specific siRNAs led to a rescue of the repressed deciliation shown in single depletion (Fig 5I and J). Together, these data indicate adequate glutamylation of ciliary tubulin, coordinated by CEP41 and CCP5, is important in the modulation of cilia disassembly.

To understand how CEP41 and CCP5 work together to regulate the glutamylation of ciliary tubulin and thereby induce EC deciliation, we measured glutamylation intensity of the cilia in each type of transfected cell under deciliation conditions (shear stress or serum retrieval). We found that ciliary glutamylation intensity is reduced by *CEP41* depletion and increased by *CCP5* depletion, but that simultaneous depletion of both genes neutralizes this effect (Fig 5H and K). The finding that *CCP5* depletion represses cilia disassembly without affecting glutamylation itself (no change in the number of GT335-marked cilia) led us to speculate that careful balance of CEP41 and CCP5 is crucial for proper levels of tubulin glutamylation rather than glutamylation initiation in the cilia. In other words, unbalanced glutamylation of ciliary tubulin can prevent the proper disassembly of EC cilia. Overall, our findings demonstrate CEP41-mediated tubulin glutamylation drives angiogenesis via mechanosignal-induced deciliation of EC.

**Figure 5.  Depletion of *CEP41* prevents shear stress-induced endothelial cell dynamics and cilia disassembly.**

A, B    Control or *CEP41* siRNA-transfected HUVECs cultivated under either static or shear stress states and examined in a wound healing assay in (A). Scale bars, 600 μm. Quantification of the extent of wound closure in (A) displayed graphically by measuring the distance between the lines at the indicated time points in (B). Data are presented as mean ± SD of three independent experiments ($n \geq 3$ scratches per experimental condition). Statistical significance was determined using the two-way ANOVA followed by Tukey's *post hoc* test (\*\*\**P* < 0.01, ns: non-significant).

C–E    An *in vitro* angiogenesis assay was performed in static or shear stress-exposed HUVECs transfected with the indicated siRNAs for 18 h in (C). Scale bars, 600 μm. Quantification of tube node number in (D) and tube length in (E) from data examined within equivalent fields of view at each time point using the ImageJ angiogenesis analyzer. Data are shown as mean ± SD of five independent experiments with $\geq 5$ tubulogenesis regions per condition. Statistical significance was determined using the two-way ANOVA followed by Tukey's *post hoc* test (\*\**P* < 0.01, \*\*\**P* < 0.001, ns: non-significant).

F–H    Control or *CEP41*-deficient HUVECs cultivated under laminar shear stress and immunostained with ARL13b-specific antibodies to measure deciliation. Representative images are indicated by rectangles with numbers and shown in magnified images in the right panels. The resulting data were compared to those of cells under static conditions in (F). Scale bars, 20 μm. Quantification of ARL13b-positive ciliary cells (G) and of relative intensity of GT335/ARL13b signals in the cilia (H) in images (F) is the results of three independent experiments with $\geq 200$ cells per condition (mean ± SD (G), median (H)). \**P* < 0.05, \*\*\**P* < 0.001, ns: non-significant (two-way ANOVA with Tukey's *post hoc* test (G) and Kruskal–Wallis test with Dunn's *post hoc* test (H)).

I–K    Control, *CEP41*-, or *CCP5*-depleted hTERT-RPE1 cells cultured under serum starvation for 48 h and fixed at 18 h after serum retrieval for immunostaining with both ARL13b- and GT335-specific antibodies in (I). Images of the results 18 h after serum addition. Representative cells are indicated by rectangles. Scale bars, 20 μm. Quantification of ARL13b-positive ciliated cells (J) and relative intensity of GT335/ARL13b signals in the cilia (K) under either serum starvation or serum retrieval (18 h) conditions are the results of three independent experiments with $\geq 200$ cells per condition (mean ± SD (J), median (K)). \*\**P* < 0.01, \*\*\**P* < 0.001, ns: non-significant (one-way ANOVA with Tukey's *post hoc* test (J) and Kruskal–Wallis test with Dunn's *post hoc* test (K)).

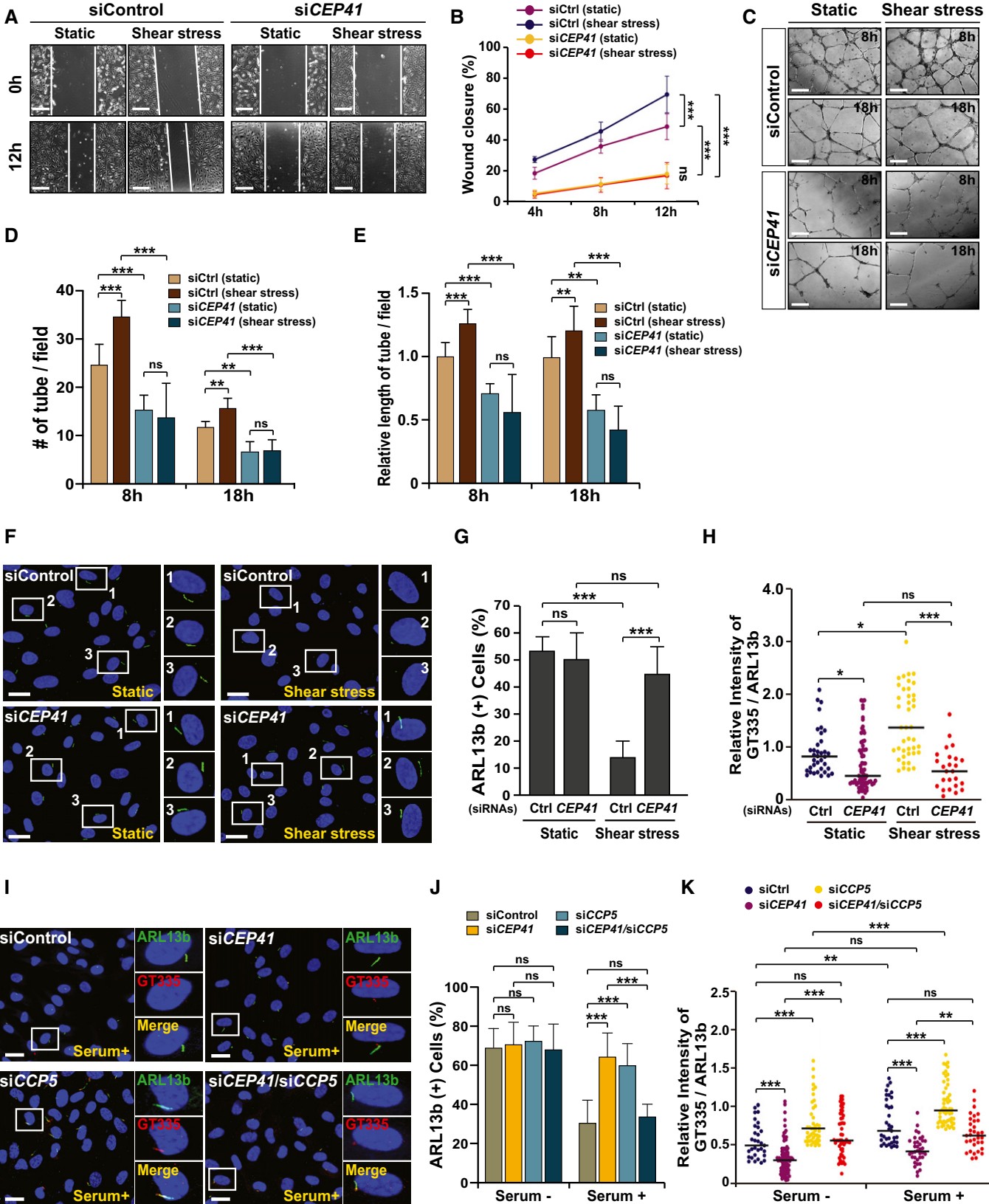

Figure 5.

## The activation of AURKA is essential for CEP41-dependent mechanotransduction to promote angiogenesis

We next asked why cilia responding to mechanical stress are disassembled in the EC dynamics that govern angiogenesis. We hypothesized that the molecular mechanisms of deciliation and angiogenesis may be linked by mechanotransduction through the endothelial cilia. To further clarify the mechanisms underlying CEP41's regulation of EC dynamics, we investigated the possible involvement of AURKA, which regulates cilia disassembly [37] and expression of pro-angiogenic genes [38,39]. First, we measured the expression of *VEGFA* and *VEGFR2*, pro-angiogenic regulators affected by shear stress [31,40], in *CEP41*-depleted HUVECs. A real-time quantitative RT–PCR (qRT–PCR) analysis revealed a greater than threefold increase in the expression of both genes in control cells exposed to shear stress. We did not observe this increase, however, in *CEP41*-depleted cells exposed to the similar shear stress (Fig 6A and B).

Next, we performed immunoblot assays with a phospho-AURKA-specific antibody to quantify AURKA activation in HUVECs and RPE1 cells under deciliation conditions (i.e., exposure to shear stress and serum starvation, respectively). We found that *CEP41*-depleted cells show reduced AURKA activation compared with control cells (Fig 6C and D and Appendix Fig S5A and B). Moreover, concurrent *CCP5* silencing restores the AURKA activation restricted by *CEP41* depletion in HUVECs under shear stress (Fig 6C). Finally, we confirmed these *in vitro* results *in vivo* using *cep41*-depleted *Tg (kdrl:eGFP)* zebrafish. We found *vegfa* and *vegfr2* mRNA levels are unaffected by *cep41* depletion at 18 hpf when the ECs of zebrafish are exposed to minimal shear stress ($\approx$ 0 dyne/cm$^2$). In contrast, *cep41*-deficient zebrafish ECs show dampened upregulation of *vegfa* and *vegfr2* mRNAs at 26 hpf when they are exposed to more significant levels of shear stress (> 0.5 dynes/cm$^2$; Fig 6E and F). In control experiments using a transgenic zebrafish with eGFP-labeled neurons, we found that non-ECs express lower *vegfa and vegfr2* than ECs at 26 hpf (Fig 6E and F). Consistent with our data from human cell lines, *cep41*-depleted zebrafish also show reduced AURKA activation in ECs exposed to shear stress (Fig 6G and H). Together, these results suggest that mechanosensation in ECs influences angiogenesis via a mechanism in which CEP41 regulates the expression of pro-angiogenic genes by activating AURKA.

Next, to determine whether the activation of AURKA is directly involved in CEP41-mediated deciliation, we measured the effects of exogenous AURKA-T288D, a constitutively active form of AURKA, on the repression of deciliation in *CEP41*-deficient cells. We found exogenous AURKA-T288D induces cilia disassembly of hTERT-RPE1 cells even in serum starvation condition, and it can rescue the repressed deciliation of *CEP41*-depleted cells in serum retrieval condition (Fig 6I and J). We also measured the effects of exogenous AURKA-K162R, an inactive form of AURKA, as a negative control, and found it cannot rescue the *CEP41* depletion-derived deciliation defects under serum retrieval (Fig 6I and J).

To further demonstrate whether shear stress-dependent activation of AURKA is associated with CEP41-mediated angiogenesis, we performed *in vitro* angiogenesis assay in the presence of exogenous AURKA-T288D or AURKA. We found in both static and shear stress conditions, exogenous AURKA-T288D enhances tubulogenesis of control EC and restores the defects of *CEP41*-depleted EC tubulogenesis (Fig 6K–N and Appendix Fig S6). In contrast, the tubulogenic capacity of *CEP41*-depleted cells is unaffected by exogenous AURKA-K162R under shear stress (Fig 6M and N, and Appendix Fig S6B). Intriguingly, we found exogenous AURKA has little effect on tubulogenesis in both control and *CEP41*-depleted ECs under static conditions, whereas their tubulogenic capacity is sufficiently enhanced by exogenous AURKA under shear stress conditions (Fig 6K–N and Appendix Fig S6). Together, these results indicate

---

**Figure 6.  CEP41 regulates angiogenesis via AURKA-activated mechanotransduction.**

A, B  *VEGFA* (A) and *VEGFR2* (B) mRNA levels quantified by qRT–PCR in control or *CEP41*-knockdown HUVECs under static or shear stress conditions. The expression of *GAPDH* was quantified for the normalization of the qRT–PCR results. Data are shown as mean $\pm$ SD of three independent experiments. Statistical significance was determined with the Brown–Forsythe ANOVA followed by Dunnett's T3 *post hoc* test (**$P < 0.01$, ***$P < 0.001$, ns: non-significant).

C, D  Immunoblot assays for phospho-AURKA and AURKA were performed in HUVECs (C) and hTERT-RPE1 cells (D) transfected with the indicated siRNAs. Protein levels were normalized against β-actin in the same blots.

E, F  The mRNA levels of zebrafish *vegfa* (E) and *vegfr2* (F) were quantified by qRT–PCR in eGFP-positive ECs of control- or *cep41*-MO-injected *Tg(kdrl:eGFP)* zebrafish at 18 hpf (low shear stress) and 26 hpf (high shear stress). The expression of *vegfa* and *vegfr2* was quantified in neuronal cells from control MO-injected zebrafish for comparisons with that of ECs. The expression of zebrafish *β-actin* was quantified for the normalization of those qRT–PCR results. Data are shown as mean $\pm$ SD of three independent experiments. Statistical significance was determined using the one-way ANOVA followed by Tukey's *post hoc* test (***$P < 0.001$, ns: non-significant).

G, H  *Tg(kdrl:eGFP)* zebrafish were injected with control- or *cep41*-MOs and subjected to immunostaining with phospho-AURKA-specific antibodies (red) and DAPI at 26 hpf. The insets indicate the representative area from each immunostaining experiment, and the red dots indicate EC AURKA activation. Scale bars, 40 μm. Quantification of the phospho-AURKA-positive ECs (H) in equivalent fields of view for each MO-injected zebrafish in images in (G) is the result of three independent experiments with $\geq$ 10 embryos per condition. The top and bottom whiskers represent the maximum and minimum values, respectively. **$P < 0.01$ (unpaired Student's *t*-test with Welch's correction).

I, J  *CEP41*-depleted hTERT-RPE1 cells were transfected with vectors encoding nothing (MOCK), AURKA-T288D, or AURKA-K162R and then immunostained with ARL13b-specific antibodies after serum starvation or serum retrieval. Images were taken from the results of staining done 18 h after serum retrieval. Rectangles indicate the representative cells that appear in the magnified images in the right panels. Scale bars, 20 μm. A quantification of ARL13b-labeled ciliated cells under the indicated conditions in (J). These are the results of three independent experiments with $\geq$ 200 cells per condition (mean $\pm$ SD). *$P < 0.05$, ***$P < 0.01$, ns: non-significant (two-way ANOVA with Tukey's *post hoc* test).

K–N  Control and *CEP41*-deficient HUVECs were transfected with expression vectors encoding nothing (MOCK), AURKA, AURKA-T288D, or AURKA-K162R and subjected to an *in vitro* angiogenesis assay for 18 h under static (K, L) or shear stress (M, N) states. Quantification of tube node number in (K, M) and tube length in (L, N) from data examined within equivalent fields of view at each time point using the ImageJ angiogenesis analyzer. Data are shown as mean $\pm$ SD of five independent experiments with $\geq$ 5 tubulogenesis regions per condition. Statistical significance was determined using the two-way ANOVA followed by Tukey's *post hoc* test (*$P < 0.05$, **$P < 0.01$, ***$P < 0.001$, ns: non-significant).

Source data are available online for this figure.

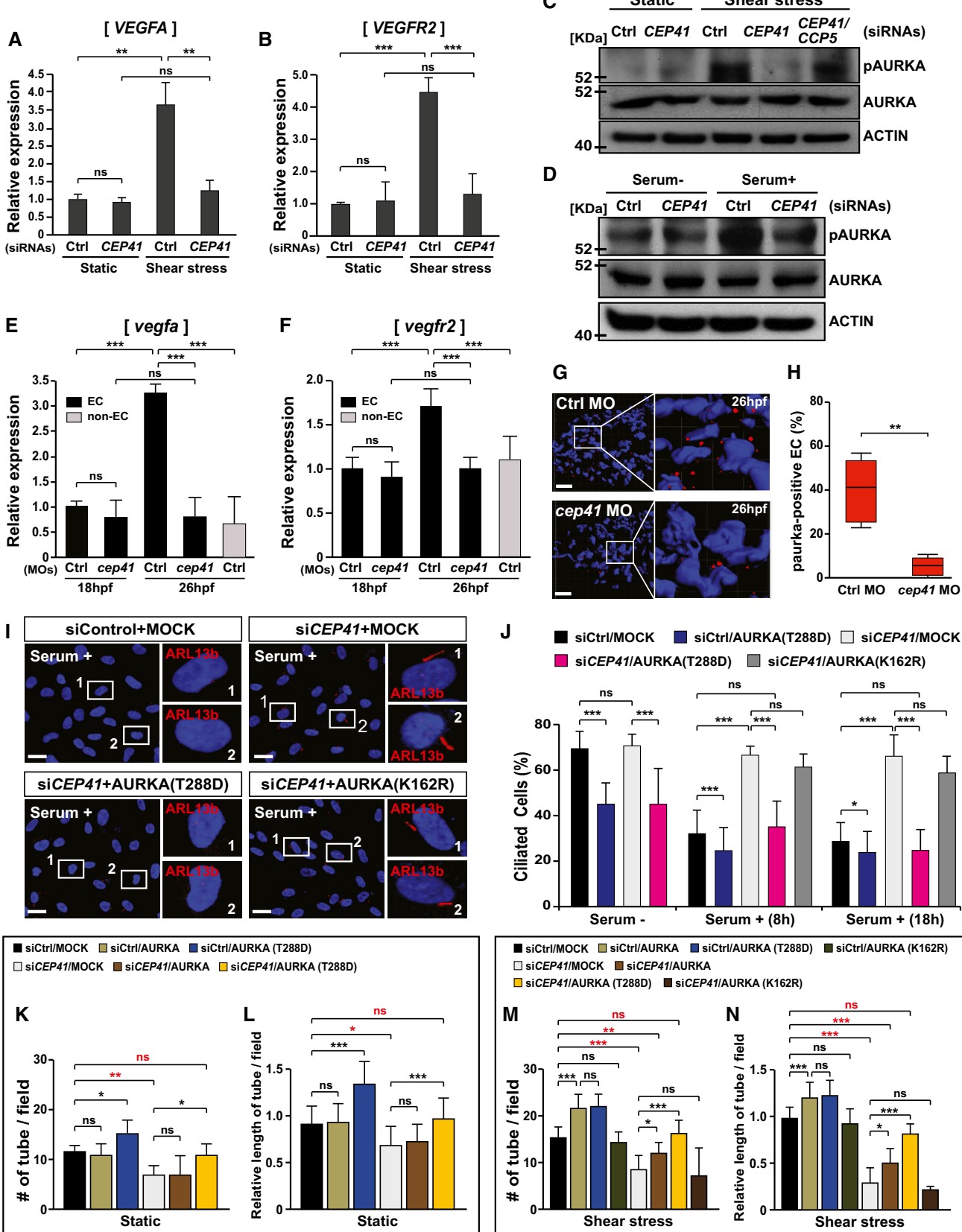

**Figure 6.**

AURKA activation is important in CEP41-mediated mechanotransduction acting on EC dynamics and deciliation to control angiogenesis.

## CEP41 stabilizes HIF1α to modulate hypoxia-induced angiogenesis via the AURKA-VEGF pathway

Under hypoxic conditions, hypoxia-inducible factor 1-alpha (HIF1α) is activated in ECs to induce the extension of existing blood vessels and the formation of new vessels [41]. Similarly, hypoxic tumor cells require activation of both AURKA and HIF1α to induce the expression of the angiogenic regulators that facilitate angiogenesis [41,42]. Our results described above led us to hypothesize that CEP41 regulates hypoxia-induced angiogenesis by activating both HIF1α and AURKA. The cilia-related role of HIF1α in ECs, however, has not been well studied with regard to angiogenesis. Thus, we first asked whether hypoxia affects deciliation and tubulin glutamylation in EC cilia, both of which are controlled by CEP41-AURKA molecular axis in the process of angiogenesis. After exposing HUVECs to hypoxia for between 1 and 4 h and staining them with both ARL13b- and GT335-specific antibodies, we compared the resulting cilia with those of cells under normoxic conditions. We found HUVECs exposed to 3 and 4 h of hypoxia show significant increases in ciliary resorption (Fig EV4A–D) and tubulin glutamylation (Fig EV4E). This implicates an involvement of HIF1α in CEP41-mediated angiogenesis in response to hypoxia.

As noted previously, hypoxia induces the expression of HIF1α and the activation of AURKA [43], which promotes cilia disassembly [44]. Accordingly, we investigated the possibility that HIF1α affects the CEP41-AURKA molecular axis to drive angiogenesis. We first found that under hypoxic conditions, control HUVECs activate AURKA; in contrast, *CEP41*-depleted HUVECs repress the activation of AURKA (Fig 7A). As expected, hypoxic HUVECs show increased mRNA levels of *VEGFA* and *VEGFR2,* which are transcriptional targets of both HIF1α [41,42] and AURKA [38,39]. However, *CEP41*-depleted cells show a minimal increase in *VEGFA* and *VEGFR2* levels under hypoxic conditions (Fig 7B and C). Consistent with the results

from human cell lines, our *in vivo* data revealed that AURKA is activated in control *Tg(kdrl:eGFP)* zebrafish ECs exposed to hypoxia; in contrast, this AURKA activation is restrained in the hypoxic *cep41*-deficient zebrafish ECs (Fig 7D and E). Moreover, we found that the hypoxia-induced upregulation of *vegfa* and *vegfr2* mRNAs is repressed by *cep41* depletion in zebrafish ECs (Fig 7F and G).

Next, we asked whether the CEP41-dependent activation of AURKA is essential for hypoxia-induced angiogenesis, as shown in shear stress-induced angiogenesis (Fig 6K–N). The HUVECs transfected with control or *CEP41* siRNAs were exposed to hypoxia and subjected to *in vitro* angiogenesis assay in the presence of exogenous AURKA-T288D or AURKA. We first confirmed that control cells enhance tubulogenesis by hypoxia exposure, while *CEP41*-silenced cells do not (Fig EV5A–C). We found exogenous AURKA hardly affects tubulogenesis in control cells and it is unable to rescue the tubulogenesis defects caused by *CEP41* depletion (Fig 7H–J). We also found that exogenous AURKA-T288D boosts tubulogenesis in *CEP41*-depleted cells, although it is insufficient to rescue to the levels of tubulogenesis observed in hypoxic control cells (Fig 7H–J). These results thus suggest that CEP41 regulates hypoxia-induced angiogenesis through AURKA activation. Next, to determine whether HIF1α mediates the CEP41-dependent AURKA activation, the effect of exogenous HIF1α on tubulogenesis in *CEP41*-depleted cells under hypoxia was examined. Similar to the result on AURKA-T288D overexpression, exogenous HIF1α rescued the tube formation defects in *CEP41*-depleted cells, but the restoration was partial (Fig 7H–J). Together, our data suggest that AURKA activation that seems to be influenced by HIF1α is essential for CEP41-dependent hypoxia-derived angiogenesis.

Based on the HIF1α dependency of CEP41-mediated angiogenesis, we hypothesized that HIF1α acts downstream of CEP41. To test this, we first used immunoblot assays to determine whether CEP41 affects the activation of HIF1α under hypoxia. We found *CEP41*-depleted cells produce less HIF1α protein than control cells (Fig 8A). Next, we asked whether CEP41 physically interacts with HIF1α to regulate its activation using co-immunoprecipitation (IP) assays. We found that our CEP41-specific antibodies were unsuitable for the co-

---

**Figure 7. CEP41 regulates hypoxia-induced angiogenesis by activating AURKA.**

A The control or *CEP41* siRNA-transfected HUVECs were cultivated under hypoxia (1% O$_2$), and their whole-cell lysates were used for immunoblot assays of CEP41, phospho-AURKA, and AURKA. The resulting data were also compared to those of normoxic cells. Protein levels were normalized against β-actin in the same blots.

B, C *VEGFA* (B) and *VEGFR2* (C) mRNA levels were measured by qRT–PCR using cDNA from normoxic and hypoxic controls or *CEP41*-depleted cells. The expression of *GAPDH* was quantified for the normalization of those qRT–PCR results. Data are shown as mean ± SD of three independent experiments. Statistical significance was determined with the one-way ANOVA followed by Tukey's *post hoc* test (***$P < 0.001$, ns: non-significant).

D, E *Tg(kdrl:eGFP)* zebrafish were injected with control- and *cep41*-MOs and then incubated under hypoxia at 28 hpf (a stage of high shear stress) for 2 h. They were subjected to immunostaining with phospho-AURKA-specific antibodies (red) and DAPI at 30 hpf. The insets indicate the representative areas from each immunostaining, and red dots indicate activated AURKA within ECs. Scale bars, 40 μm. Quantification of phospho-AURKA-positive ECs (E) in equivalent fields of view for each MO-injected zebrafish in (D) is the result of three independent experiments with ≥ 20 embryos per condition. The top and bottom whiskers represent the maximum and minimum values, respectively. *$P < 0.05$ (unpaired Student's *t*-test).

F, G Zebrafish *vegfa* (F) and *vegfr2* (G) mRNA levels were quantified by qRT–PCR in eGFP-positive ECs of control- or *cep41*-MO-injected *Tg(kdrl:eGFP)* zebrafish subjected to either normoxia or hypoxia at 30 hpf. The expression of zebrafish *β-actin* was quantified for the normalization of these qRT–PCR results. Data are shown as mean ± SD of three independent experiments. Statistical significance was determined using the one-way ANOVA followed by Tukey's *post hoc* test (*$P < 0.05$, ***$P < 0.001$, ns: non-significant).

H–J Control and *CEP41*-deficient HUVECs, transfected with expression vectors encoding nothing (MOCK), AURKA, AURKA-T288D, HIF1α, or HIF1α with CEP41, were cultivated under hypoxia and subjected to an *in vitro* angiogenesis assay for 18 h. Scale bars, 400 μm. Quantification of tube node number in (I) and tube length in (J) from data examined within equivalent fields of view at each time point using the ImageJ angiogenesis analyzer. Data are shown as mean ± SD of five independent experiments with ≥ 5 tubulogenesis regions per condition. Statistical significance was determined using the two-way ANOVA followed by Tukey's *post hoc* test (*$P < 0.05$, ***$P < 0.001$, ns: non-significant).

Source data are available online for this figure.

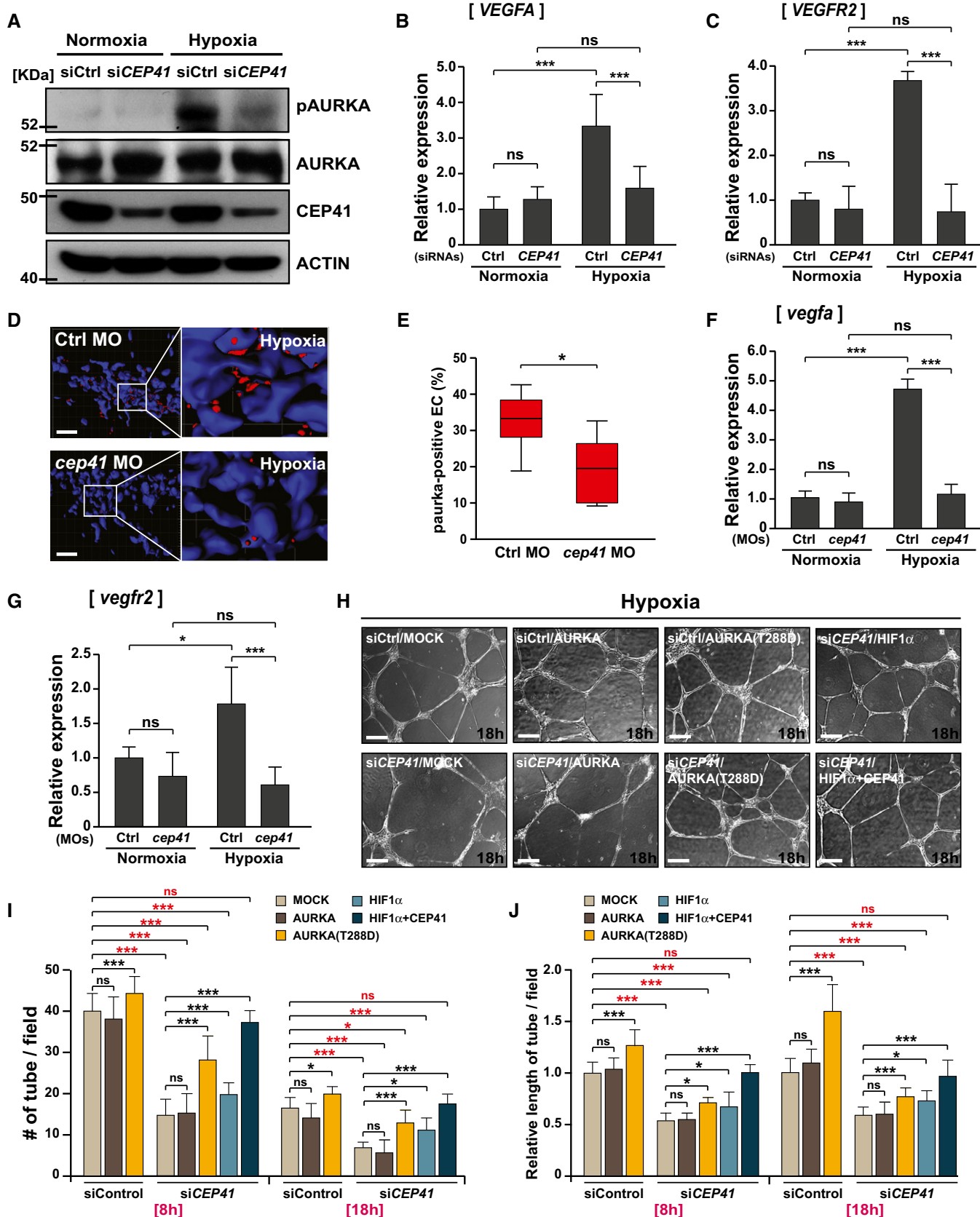

**Figure 7.**

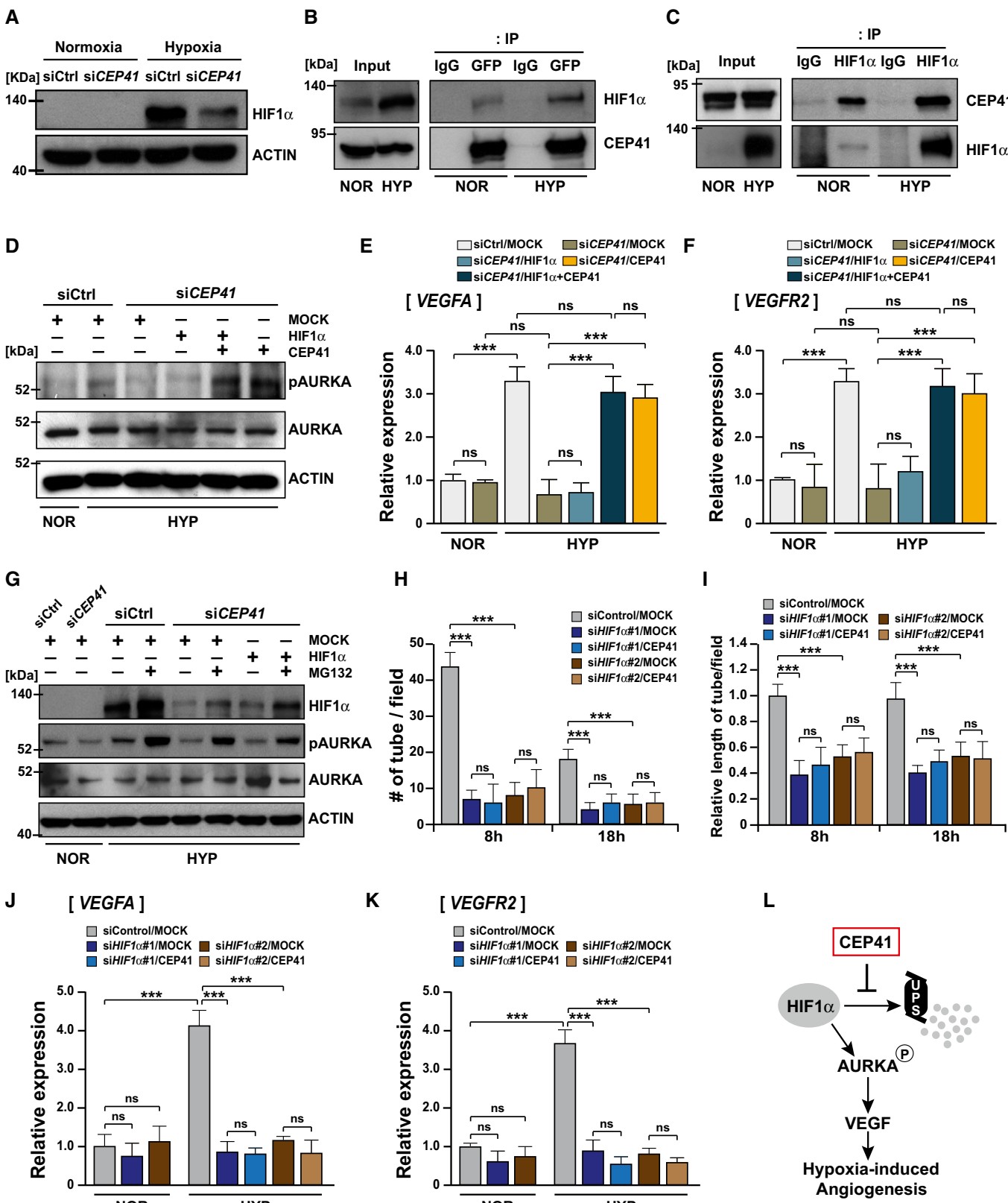

Figure 8.

**Figure 8. CEP41 functions upstream of HIF1α through physical interaction with it to drive hypoxia-induced angiogenesis.**

A HUVECs transfected with control or *CEP41* siRNAs were cultivated under hypoxia, and their whole-cell lysates were used for immunoblot assays of HIF1α. The HIF1α expression of hypoxic cells was compared to that of normoxic cells, and the protein levels were normalized against β-actin in the same blot.

B, C Co-immunoprecipitation assays using HIF1α- and CEP41-specific antibodies were performed in HEK 293T cells expressing GFP-CEP41 under normoxia or hypoxia. An IP using IgG-specific antibodies was performed as a negative control. NOR, normoxia; HYP, hypoxia.

D The *CEP41*-depleted HUVECs were transfected with expression vectors encoding nothing (MOCK), CEP41, or HIF1α and cultivated under hypoxia. Whole-cell lysates were used for immunoblot assays for phospho-AURKA and AURKA, and the resulting data were compared to those of normoxic and hypoxic control cells. Protein levels were normalized against β-actin in the same blots. NOR, normoxia; HYP, hypoxia.

E, F The *CEP41*-deficient HUVECs were transfected with expression vectors encoding nothing (MOCK), CEP41, HIF1α, or HIF1α with CEP41 and exposed to hypoxia. *VEGFA* (E) and *VEGFR2* (F) mRNA levels were analyzed by qRT–PCR using cDNA from transfected cells. Data are shown as mean ± SD of three independent experiments. Statistical significance was determined using the one-way ANOVA followed by Tukey's *post hoc* test (***$P < 0.001$, ns: non-significant). NOR, normoxia; HYP, hypoxia.

G Control or *CEP41*-depleted HUVECs were transfected with expression vectors encoding nothing (MOCK) or HIF1α and treated with MG132 or not, and then, the cells were cultivated under hypoxia. Whole-cell lysates were used for immunoblot assays for HIF1α, phospho-AURKA, and AURKA, and the protein levels were normalized against β-actin in the same blots. NOR, normoxia; HYP, hypoxia.

H, I Quantification of tube node number in (H) and tube length in (I) from *in vitro* angiogenesis assays with *HIF1α* #1 or #2 siRNAs-transfected HUVECs overexpressing nothing (MOCK) or CEP41 under hypoxia. Data are shown as mean ± SD of five independent experiments with ≥ 5 tubulogenesis regions per condition. Statistical significance was determined with the two-way ANOVA followed by Tukey's *post hoc* test (***$P < 0.001$, ns: non-significant).

J, K *VEGFA* (J) and *VEGFR2* (K) mRNA levels were analyzed by qRT–PCR using cDNA from hypoxic *HIF1α*-depleted HUVECs transfected with control or CEP41 expression vectors. The results were compared to those of control and normoxic *HIF1α*-depleted cells. Data are shown as mean ± SD of three independent experiments. Statistical significance was determined using the one-way ANOVA followed by Tukey's *post hoc* test (***$P < 0.001$, ns: non-significant).

L A schematic diagram showing the molecular mechanism by which CEP41 drives hypoxia-induced angiogenesis.

Source data are available online for this figure.

IP assays in HUVECs. Instead, we performed these assays in HEK 293T cells and observed CEP41 binds HIF1α under both normoxic and hypoxic conditions. Of note, we found hypoxia induces more expression of HIF1α and increases its binding to CEP41 (Fig 8B and C). Hence, these results suggest CEP41 modulates the activation of HIF1α via a physical interaction.

To clarify whether CEP41-dependent HIF1α activation affects the activation of AURKA under hypoxia, we examined the influences of exogenous HIF1α in hypoxic *CEP41*-depleted HUVECs by immunoblot assays for pAURKA. Similar to our results in Fig 7H–J, we found exogenous HIF1α cannot sufficiently restore the repressed activation of AURKA in the absence of *CEP41*. Instead, the cells co-expressing exogenous CEP41 show a full recovery of AURKA activation (Fig 8D and Appendix Fig S5F). Consistent with this, we found exogenous HIF1α and CEP41 together but not exogenous HIF1α alone can rescue the reduced expression of *VEGFA/R2* mRNAs in *CEP41*-depleted HUVECs under hypoxic conditions (Fig 8E and F). Notably, we observed that exogenous CEP41 rescues the suppressed AURKA activation and *VEGFA/R2* expression of hypoxic *CEP41*-depleted cells as much as the levels of hypoxic control cells (Fig 8D–F and Appendix Fig S5F). These results prompted us to hypothesize that CEP41 is involved in the stabilization of HIF1α; CEP41 may be required for the prevention of HIF1α degradation by the ubiquitin–proteasome system (UPS) in hypoxic conditions. To assess the validity of this hypothesis, we measured AURKA activation in *CEP41*-depleted HUVECs after introducing exogenous HIF1α while inhibiting the UPS. We found that *CEP41*-depleted cells over-expressing HIF1α show increased HIF1α protein and AURKA activation only when the UPS is inhibited (Fig 8G and Appendix FigS5G and H). Overall, these data indicate CEP41 is essential to prevent UPS-dependent degradation of HIF1α inducing the AURKA activation and the *VEGFA/R2* upregulation expression under hypoxia.

Next, to ensure that HIF1α regulates the hypoxia-induced angiogenesis by acting as a downstream mediator of CEP41, we examined the effects of exogenous CEP41 on the angiogenic defects caused by *HIF1α* deficiency. After transfecting HUVECs with control or

validated *HIF1α* siRNAs (Appendix Fig S7) along with empty vector or *CEP41* plasmids, we subjected them to *in vitro* tube formation assays under hypoxic conditions. We found *HIF1α*-depleted HUVECs show profoundly impaired tubulogenesis, with fewer tube nodes and a shorter cumulative tube length (Fig 8H and I, and Appendix Fig S8). Expectedly, exogenous CEP41 was unable to rescue the angiogenic defects of *HIF1α*-depleted HUVECs under hypoxia (Fig 8H and I, and Appendix Fig S8). In line with these data, exogenous CEP41 failed to restore the repressed upregulation of *VEGFA/R2* mRNAs in *HIF1α*-depleted cells under hypoxia (Fig 8J and K). Considering these data with those in Fig 7, we propose that CEP41 triggers the AURKA-VEGF pathway via the stabilization of HIF1α to modulate hypoxia-induced angiogenesis (Fig 8L).

The autocrine effect of HIF1α-induced *VEGF* expression in ECs is important for tumor angiogenesis [45]. We thus asked whether under cancer cell-induced hypoxic conditions, CEP41 influences EC dynamics to induce tumor angiogenesis *in vivo*. First, we cultured several kinds of tumor cells derived from human breast and mouse lung cancers. Next, we engrafted these cells into the perivitelline space of control MO- or *cep41* MO-injected zebrafish after color labeling (Appendix Fig S9A and B). We then observed the development of sprouting subintestinal veins (SIVs) in each type of zebra-fish under hypoxia conditions induced by the growing tumor cells. By counting the number of SIV branches and measuring the length of protruding SIVs, we found that while control zebrafish show the significant venous sprouting, *cep41*-deficient zebrafish do not show the formation of SIV sprouts (Appendix Fig S9C–E). These *in vivo* results indicate an essential role for CEP41 in tumor angiogenesis that is likely attributable to its activation of HIF1α in ECs under hypoxic conditions.

## Discussion

Here, we report that glutamylation of ciliary axonemal tubulin controls EC dynamics to promote angiogenesis. We also present

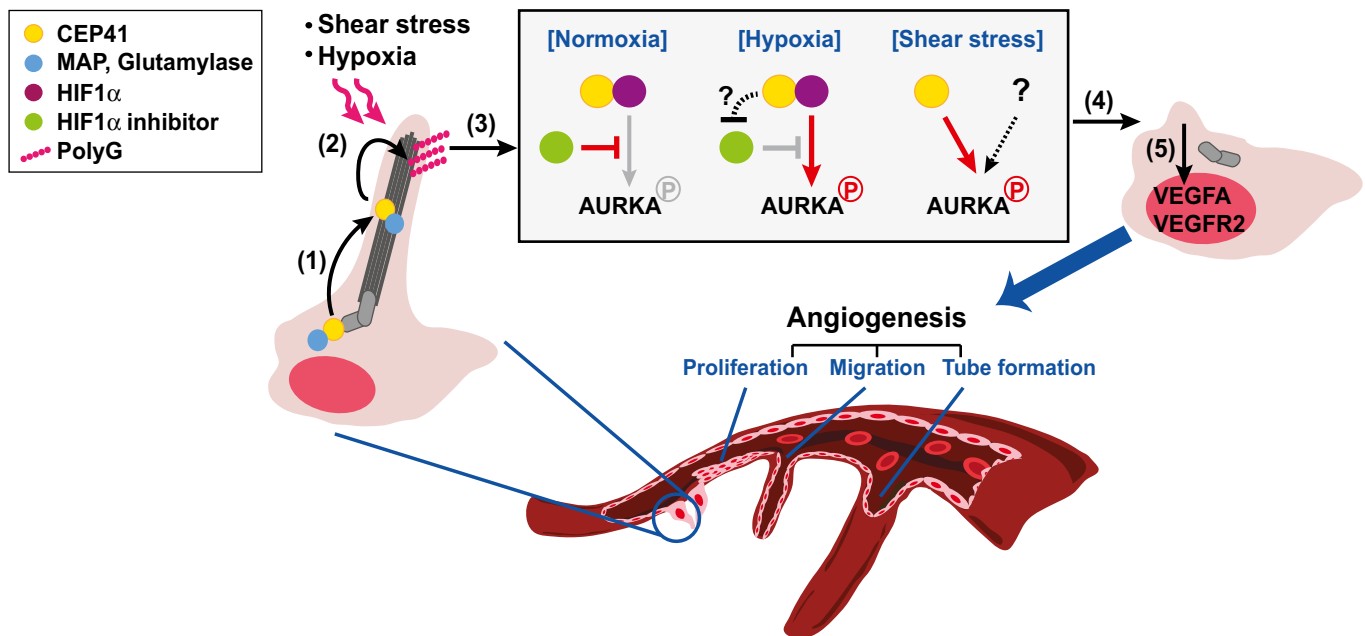

**Figure 9. Working model.**

Overall proposed model for CEP41-mediated angiogenesis *via* endothelial cilia. (1) Potential interactions of CEP41 with MAP and/or polyglutamylase and their transportation from the centrosome to the cilium, (2) tubulin glutamylation in endothelial cilia by fluid flow-driven shear stress or hypoxia, (3) activation of AURKA dependent on CEP41 under shear stress or hypoxia, (4) induction of cilia disassembly via AURKA activation, and (5) upregulation of pro-angiogenic regulators, such as *VEGFA* and *VEGFR2*.

CEP41 as a novel regulator of angiogenesis. We previously reported that CEP41 coordinates ciliary tubulin glutamylation by its physical interaction with TTLL6, a tubulin polyglutamylase [21]. In this study, we show that tubulin glutamylation in endothelial cilia is essential for the activation of AURKA downstream of fluid flow-induced mechanosensation. This, in turn, promotes cilia disassembly and the upregulation of pro-angiogenic regulators, such as *VEGFA* and *VEGFR2* both *in vitro* and *in vivo*. We further show that the CEP41-AURKA-VEGF pathway functions in hypoxia-induced angiogenesis via the stabilization of HIF1α. These findings suggest that CEP41 is a prime determinant of coordinating the transduction of mechanical stimuli in endothelial cilia through its role in tubulin glutamylation in the regulation of angiogenesis (Fig 9).

Microtubule dynamics are generally regulated by plus-end tracking proteins that directly bind to growing microtubule tips in a PTM-dependent manner [46]. Several tubulin PTMs occur during vascular EC polarization and migration, crucial events in the early stages of angiogenesis [8–10,47]. Our understanding of the mechanisms by which PTMs govern angiogenesis, however, has been limited to the (de)acetylation of cytoplasmic tubulin [22,48]. Previous studies that identified microtubule-associated proteins (MAPs) as regulators of angiogenesis [48,49] have supported the essential nature of microtubule dynamics in angiogenesis [3]. Among several MAPs, tracking proteins such as end-binding proteins (EBs) and cytoplasmic linker-associated proteins (CLASPs) are of particular interest, because they are located at both the plus and minus microtubule ends where they can control various cellular behaviors (i.e., polarity, migration, and differentiation) that rely on tubulin PTMs [50,51]. Accordingly, in the future, to better understand the

mechanisms by which CEP41-mediated ciliary tubulin glutamylation controls EC dynamics, we will further investigate whether CEP41 binds directly to microtubules. We will also assess the involvement of other MAPs like EB1, which are already associated with cilia-dependent angiogenesis.

Of note, *CCP5* depletion in both human cell lines and zebrafish attenuates angiogenesis like *CEP41* depletion, but simultaneous depletion of *CCP5* and *CEP41* rescues their similar angiogenesis defects. Moreover, we found that each of these single knockdowns affects the levels of ciliary glutamylation in opposite ways. Remarkably, we found a correlation between ciliary length and the extent of ciliary glutamylation under various cilia assembly states. More intriguingly, in cilia disassembly conditions, the depletion of either *CCP5* or *CEP41* alone similarly represses deciliation despite their contrasting effects on ciliary glutamylation. Given previous reports [27,52], we speculate that the opposing effects CEP41 and CCP5 have on ciliary glutamylation may be due to reciprocal inhibition. If glutamylation in cilia depends on a functional balance between these two molecules, dysregulation of either one may disrupt appropriate ciliary glutamylation to affect ciliary length and cilia disassembly. Although CCP5 reportedly functions as a deglutamylase that inhibits the elongation of glutamate chains [27], CEP41's role in this process is unclear. One possible scenario comes from our previous finding that CEP41 interacts with TTLL6, a polyglutamylase that elongates glutamate chains [21]. It remains conceivable, however, that CEP41 binds other TTLLs that act in glutamylation initiation. Thus, to clarify how the functions of CEP41 and CCP5 are coordinated to regulate ciliary tubulin glutamylation in angiogenesis, we plan to identify CEP41's interactions with other TTLLs and CCPs

and determine how the ciliation/deciliation are affected by the ciliary glutamylation they regulate.

Consistent with the general role of cilia as cellular antenna, previous studies have suggested that vascular endothelial cilia sense and transduce mechanical shear stress stimuli induced by blood flow [12–16,28]. Thus, the loss of cilia themselves or a dysregulation of tubulin acetylation that affects ciliary assembly would lead to EC dysfunction during angiogenesis [15,22]. Here, we show that CEP41-mediated ciliary tubulin glutamylation is essential for shear stress-responded EC dynamics, while not essential for an initial formation of cilia. Moreover, we demonstrate that cilia disassembly rather than cilia assembly is involved in CEP41-mediated EC dynamics under shear stress. This is consistent with previous reports that tubulin glutamylation causes microtubule severing [53] and that high shear stress promotes resorption of endothelial cilia [36]. We further show CEP41 activates AURKA and induces expression of *VEGFA* and *VEGFR2* in ECs under shear stress. Thus, we suggest that the CEP41-AURKA-VEGF pathway through endothelial cilia is critical for the control of shear stress-induced angiogenesis. Although our data imply ciliary tubulin glutamylation is responsible for AURKA-mediated deciliation and expression of *VEGFA/R2*, it remains unclear whether cilia disassembly is directly associated with the transcriptional regulation of *VEGFA/R2* expression. Thus, this will require further investigations to determine the mechanism linking these two biological events. Moreover, we found that only under shear stress, exogenous AURKA can induce a limited recovery from *CEP41* depletion-induced defects in angiogenesis, although shear stress itself minimally increases the levels of endogenous *VEGFA/R2* mRNAs and activated AURKA proteins in the absence of *CEP41* (Fig 6A–C, M, N, and Appendix Fig S5A). This led us to speculate that (i) shear stress can somehow activate exogenous AURKA independently of CEP41 ((3) in Fig 9) and (ii) this CEP41-independent mechanism is involved in the regulation of angiogenesis via other pro-angiogenic regulators like cytokines. Thus, future studies will be necessary to identify the molecular mechanisms that promote shear stress-induced, CEP41-independent angiogenesis, as well as how these mechanisms relate to the CEP41-dependent mechanisms.

As noted, tumors induce angiogenesis not only by expressing pro-angiogenic genes [41,42] but also by activating nearby ECs [54]. We here find that CEP41 facilitates hypoxia-induced angiogenesis through HIF1α-AURKA-VEGF pathway *in vitro* and further show that CEP41 affects EC dynamics under tumor-induced hypoxic conditions *in vivo*. It, however, is unclear whether HIF1α is directly involved in the tubulin glutamylation of cilia and the cilia-associated modulation in EC dynamics. Nevertheless, it is possible that CEP41 regulates angiogenesis by cilia disassembly via HIF1α, on the basis of previous studies that *HIF1α* depletion inhibits AURKA-dependent cilia disassembly [44,55,56] and our findings that hypoxia induces EC deciliation associated with tubulin glutamylation (Fig EV4). In addition, our data suggest CEP41 is responsible for preventing HIF1α degradation to modulate AURKA-mediated EC dynamics in the hypoxia-induced angiogenesis. Thus, further studies will be required to identify the molecular mechanisms by which CEP41 stabilizes HIF1α. We expect that this occurs via the repression of a specific HIF1α inhibitor, leading to the degradation of HIF1α under hypoxic conditions ((3) in Fig 9). Finally, our findings suggesting an essential role of CEP41 in tumor angiogenesis provide novel insight into the development of cancer therapeutics targeting CEP41 or ciliary tubulin glutamylation.

# Materials and Methods

### Cell culture

Human umbilical vein endothelial cells (HUVECs) were purchased from Lonza (Walkersville, MD, USA) and grown at 37°C and 5% $CO_2$ in EGM™ BulletKit™ medium (Lonza, CC-3162: EBM2 + FBS, Hydrocortisone, hFGF, VEGF, R3-IGF, Ascorbic acid, hEGF, Heparin, GA-1000) according to the manufacturer's recommendation. The HUVECs were passaged by trypsinization (0.25%, Corning, 25-053-CI, NY, USA), and only passages 3–5 were used for experiments. Human melanoma MDA-MB-435 and human embryonic kidney 293T (HEK 293T) cell lines were grown in DMEM (Corning, 10-013-CVR); human TERT-Retinal Pigment Epithelium 1 (RPE1) cells were cultured in DMEM F-12 (Corning, 10-090-CVR); and human ductal breast epithelial tumor cell line T47D and mouse primary lung epithelial tumor cell line TC-1 were maintained in RPMI 1640 (corning, 15-040-CVR) at 37°C and 5% $CO_2$. All of cell culture media were supplemented with 10% fetal bovine serum (FBS) (Corning, 35-016-CV) and 1% penicillin–streptomycin (PS) (Corning, 30-002-CI).

### Transfection and treatment

For siRNA transfection of HUVECs, $2 \times 10^4$ cells/cm$^2$ were plated per well the day before transfection. When the cells had reached approximately 40–50% confluency, they were washed with HEPES (Lonza, CC-5024) to remove residual antibiotics and then transfected with 30 nM siRNAs non-targeting and targeting *CEP41*, *CCP5* (GE Dharmacon, ON-TARGET *plus* siRNA reagents, CO, USA, and Bioneer, Korea), or *HIF1α* (Bioneer, Korea) using Lipofectamine RNAimax (Invitrogen, 13778-150, CA, USA) in the transfection medium (EGM™ BulletKit™ medium without GA-1000). The following day, cells were washed with HEPES and the culture medium was changed with the complete medium containing GA-1000. For siRNA transfection of hTERT-RPE1 cells, $3 \times 10^4$ cells/cm$^2$ were resuspended in DMEM/F12 supplemented with 10% FBS and seeded on Lab-Tek 8-well chamber slides (Nunc, 154534, NY, USA). About 60–70% confluent cells were washed with DPBS (Corning, 21-030-CVR) and transfected with 30 nM siRNAs using Lipofectamine RNAimax in Transfectagro medium (Corning, 40-300-CVR). For plasmid DNA transfection of HUVECs, human CEP41 coding sequences were subcloned into pIC113 vector or pcDNA3.1 vector. Human AURKA coding sequences were cloned and mutagenized to generate AURKA (T288D) and AURKA (K162R) using a QuickChange II Site-Directed Mutagenesis Kit (Agilent Technologies, #200524). The subcloned DNAs (0.5–1 ng/μl) were transfected into HUVECs or HEK293T cells using Lipofectamine 3000® (Invitrogen, L3000015). The sequences of siRNAs are in Appendix Table S1. Proteasome inhibitor MG132 (Calbiochem, 474790, CA, USA) was dissolved in dimethyl sulfoxide (DMSO, Corning, 25-950-CQC) at 20 mM as a stock solution. Transfected cells were treated with 0.7 nM MG132 for 3 h.

## Real-time quantitative RT–PCR

RNAs were extracted from the transfected HUVEC cells with RNeasy Mini Kit (QIAGEN, 74104, CA, USA) following the manufacturer's instructions. Total RNAs were reverse-transcribed to generate cDNA using the AccuPower® RocketScript™ RT Premix (Bioneer, K-2101, Korea). The mRNA expression levels of the target genes were measured using SYBR green (Bioneer, K-6251, Korea). The cycling parameters were as follows: 95°C for 10 min, followed by 40 cycles of 15 s at 95°C, 30 s at 60°C, and 30 s at 72°C. The mRNA expression level of each gene was normalized with the value of *GAPDH* mRNA expression. The sequences of primers (Bioneer, Korea) used for the RT–PCR are in Appendix Table S2.

## *In vitro* wound healing assay

Human umbilical vein endothelial cells transfected with siRNAs and/or plasmid DNAs were seeded onto 24-well plates overnight, and confluent monolayers of the cells were scratched using a 1000 µl pipette tip to generate the wound. Cells were washed with HEPES, and EGM™ BulletKit™ medium was added to allow wound healing. To determine the extent of wound closure, photographs of the wound were taken 0, 4, 8, and 12 h later using a phase-contrast microscope (Olympus, CKX41, Japan). Wound closure was measured proximate cells length using ImageJ software (NIH, USA).

## Transwell cell invasion assay

The transfected HUVECs with control or *CEP41* siRNAs suspended in serum-free medium were added to the upper chamber of a 3.0-µm polyester filter insert (Corning, #3472) pre-coated with 200 µl Matrigel diluted 1:3 in serum-free medium. The upper chamber was placed in a 24-well plate containing complete medium and incubated at 37°C. After 18 h, cells on the inside of the transwell insert were swabbed, and cells on the underside of the insert were fixed with 100% methanol for 10 min and stained with 0.05% crystal violet solution (Mentos, GM4455, Germany) for 20 min. Five randomly selected fields were photographed, and the number of migrated cells in the fields of view (682 × 512 pixels) was counted.

## *In vitro* angiogenesis (endothelial tube formation) assay

Human umbilical vein endothelial cells transfected with siRNAs and/or plasmid DNAs were plated onto 24-well plates pre-coated with 250 µl Matrigel (Corning, 354234) and incubated at 37°C and 5% $CO_2$. Tube formation was visualized using a light microscope (Olympus, CKX41), and then, photographs were taken 8 and 18 h later. The extent of tube formation was quantified by counting the number and total length of the formed tubes using ImageJ software (NIH, USA).

## Immunocytochemistry

Transfected cells were fixed in 100% methanol at −20°C for 10 min. After washing with 1× PBS, the cells were blocked with blocking solution (4% donkey serum in PBST) at RT for 1 h. Cells were incubated with primary antibodies: mouse anti-acetylated α

tubulin (Sigma, T7451, MO, USA, 1:2,000), mouse anti-GT335 (Adipogen, AG-20B-0020, CA, USA, 1:2,000), rabbit anti-γ tubulin (Santa Cruz, SC-10732, CA, USA 1:200), rabbit anti-ARL13b (Proteintech, 17711-1-AP, IL, USA, 1:1,000) antibodies at 4°C overnight in blocking solution. After washing with 1× PBS, the cells were incubated with Alexa Flour® 488-conjugated secondary antibodies (Invitrogen, mouse A11001, rabbit A11008, 1:500) or Alexa Flour® 594-conjugated secondary antibodies (Invitrogen, mouse A11005, rabbit A11012, 1:500) at RT for 1 h. Randomly selected fields (≥ 5) were photographed using the confocal microscope (Carl Zeiss, LSM 700 or LSM 780, Germany) in all of imaging analyses. The intensity and length of cilia were measured using ImageJ software (NIH, USA).

## Immunoblotting and Co-Immunoprecipitation

Cells were lysed with a RIPA buffer containing 1% NP-40, 150 mM NaCl, 50 mM Tris (pH 8.0), 0.5% sodium deoxycholate, 0.5 M EDTA, and 0.1% SDS. Proteins (30–50 µg/lane) were separated by SDS–PAGE and transferred onto 0.45 µm PVDF membrane (Millipore, IPVH00010, USA). After blocking in TBST (Tris-buffered saline pH 7.5 with 0.1% Tween-20) including 5% skim milk (BD Biosciences, Cat.232100, USA) or 5% normal goat serum (Millipore, S26, USA) for 1 h at room temperature, membranes were blotted with a mouse anti-HIF1α (BD Transduction Laboratories, 610958, CA, USA), mouse anti-Aurora A (Cell Signaling, #12100, MA, USA), rabbit anti-phospho T288 Aurora A (Cell Signaling, #3029, USA), mouse anti-actin (Santa Cruz Biotechnology, sc-8432, CA, USA), and rabbit anti-CEP41 antibodies overnight at 4°C. The membranes were washed three times with TBST and incubated with secondary antibodies: a goat anti-mouse IgG horseradish peroxidase (HRP)-conjugated antibody (GeneTex, GTX213111-01, CA, USA) or a goat anti-rabbit IgG HRP-conjugated antibody (GeneTex, GTX213110-01) for 1 h at room temperature. The membranes were washed three times with TBST and enhanced with chemiluminescence substrate (PerkinElmer, NEL104001EA, MA, USA) to visualize specific proteins. The expressions of protein were normalized with actin expression in each blot and quantified using ImageJ software (NIH, USA). For Co-Immunoprecipitation, the 500 µg cell lysates were incubated with 2 µg primary antibodies: mouse anti-HIF1α, rabbit anti-CEP41, mouse anti-IgG, or rabbit anti-IgG and 50 µl Dynabeads Protein G (Thermo Fisher Scientific, 10003D, MA, USA) overnight at 4°C. The beads were washed 3 times with PBST (0.5% Tween-20 in PBS) and denaturized with 50 µl of 2× sample buffer by boiling for 10 min. The immunoprecipitated samples were separated by 8% SDS–polyacrylamide gel electrophoresis.

## Shear stress modeling

Human umbilical vein endothelial cells were seeded in the special culture dishes, slightly modified from the previous protocol [29]; the bottoms of 35 mm dishes were bonded into the center of 100 mm dishes with URO-BOND®IV (Urocare Products Inc., # 5004015, CA, USA). Confluent monolayers of HUVECs were subjected to laminar shear stress for 48 h at 37°C using a $CO_2$-resistant orbital shaker (Thermo Scientific, #88881102, MA, USA) inside an incubator. On the basis of the previous description of this system

[29], a shear stress of ranging between 3.96 and 19 dynes/cm$^2$ across each monolayer was achieved at a rotating frequency of 77 and 226 rpm. HUVECs, which were not subjected to the shear stress, were cultured in the same $CO_2$ incubator and served as the control of static state.

### Induction of hypoxia

Hypoxic atmosphere (1% oxygen) was established in an Invivo2 400 hypoxia workstation (Ruskinn, UK) for 1–16 h. The HUVEC cells were exposed for the hypoxic state 48–72 h after transfection, and cells in the normoxic state were prepared as controls for each experiment. The cell extracts for RNA and protein preparation were taken inside the workstation to avoid re-oxygenation.

### Zebrafish housing and manipulations

Adult zebrafish were maintained with a cycle of 13-h light and 11-h dark in the automatic system (Genomic-Design, Korea) at 28.5°C and pH 7.0-7.9. The zebrafish embryos were collected by natural breeding and incubated in clean Petri dishes containing E3 medium (297.7 mM NaCl, 10.7 mM KCl, 26.1 mM CaCl$_2$, and 24.1 mM MgCl$_2$) containing 1% methylene blue (Samchun Chemicals, M2662, Korea) at 28.5°C. To inhibit formation of melanin, the zebrafish larvae were raised in the E3 medium containing 0.2 mM N-phenylthiourea (PTU; Sigma-Aldrich, p7629, USA). Animal subject research was reviewed and approved by the Institutional Animal Care and Use Committee at the Samsung Biomedical Research Institute and the Sungkyunkwan University.

### CRISPR/Cas9 genome editing-mediated generation of *cep41* mutants

pT7-gRNA vector was presented by Dr. Seong-Kyu Choe (Wonkwang University, Korea). The gRNA (5′-GGAAGATGTCGGTGAAG AGG-3′) was designed to target exon 1 of *cep41* using CRISPR design (http://zifit.partners.org/ZiFiT). Oligonucleotides (5′-TAGGAAGAT GTCGGTGAAGAGG-3′, 5′-AAACCCTCTTCACCGACATCTT-3′) were annealed in a thermal cycler at 95°C for 5 min followed by a slow cooling procedure: cooling to 50°C (0.1°C/s), pausing at 50°C for 10 min, and cooling to 4°C. The annealed oligonucleotides were cloned into the gRNA plasmid between BsmbI sites. To make gRNA, the template DNA was linearized by BamH1 digestion and purified using a column (GeneAll, 112-102, Korea). The gRNA was generated by *in vitro* transcription using a MEGAshortscript T7 Kit (Invitrogen, AM1354, USA). One microgram of Cas9 protein (ToolGen, TGEN_CP1, Korea) and 800 ng/μl gRNA were mixed, incubated at 37°C for 15 min, and injected in *Tg(kdrl:eGFP)* zebrafish embryos at 1–2 cell stage (1 nl of mixture). The injected embryos were raised to adulthood. F0 fish were crossed to wild-type fish (AB strain) to generate F1 progeny. F1 fish were genotyped by genomic PCR and DNA sequencing. Mutant alleles were identified by high-resolution melt analysis of PCR products generated with the following primers: F: 5′-GCGACCCATTGAGTTTTGAT-3′, R: 5′-TTAAAGCACTGACCGC TGAA-3′. Heterozygous F1 fish were crossed to generate F2 fish. The F2 fish were genotyped, and the stable lines were maintained by inbred crossing. F2 fish embryos were used for phenotypic analysis.

### Microinjection of zebrafish

To block expression of specific zebrafish genes, splicing-blocking or translation-blocking antisense morpholino oligonucleotides (MOs) were designed and synthesized by GeneTools (Philomath, OR, USA). The MOs were dissolved in nuclease-free water and microinjected into zebrafish embryos at one to two cell stages using a gas-used microinjection system (World Precision Instruments, PV83 Pneumatic PicoPump, SYS-PV830, FL, USA). The injected embryos were incubated at 28.5°C in 1× E3 medium and treated with 0.2 mM PTU at 12 h post-fertilization (hpf). The knockdown efficiency of *cep41* splicing-blocking MOs was validated by RT–PCR with designed sets of primers (Bioneer, Korea). The sequences of MOs and primers are in Appendix Table S3.

### Immunohistochemistry and imaging of zebrafish

For immunostaining of *Tg(kdrl:EGFP)* zebrafish, dechorionated embryos were collected at 28 hpf and fixed in 4% PFA at 4°C overnight. After washing with 1× PBS, the embryos were blocked in blocking solution (0.5% Tween-20 and 10% normal goat serum in PBS) at RT for 1 h. The embryos were incubated with primary antibodies (GT335, 1:100; ARL13b, 1:400; acetylated-α tubulin, 1:400; phospho T288 Aurora A, 1:100) at 4°C overnight in blocking solution. After washing with PBST (0.5% Triton X-100 in 1× PBS), the embryos were incubated Alexa Flour® 594-conjugated secondary antibodies (1: 500) at RT for 2 h. The zebrafish embryos were finally washed with PBST and mounted in 75% glycerol for imaging. The confocal images were acquired using LSM 780 laser scanning microscopy (Carl Zeiss, Germany) and analyzed using Zeiss ZEN imaging software (Germany). 3D reconstruction images of the Z-stacked images were generated using Imaris software v.9.3. (Bitplane, Zurich, Switzerland). For live imaging with MO-injected *Tg(kdrl:EGFP)* zebrafish embryos, they were anesthetized with 0.02% tricaine (Sigma, A5040, USA) and photographed in the low-melting-point agarose (Mentos, M2070, Germany) using LSM 700 confocal microscopy. The bright-field images were acquired by tile scan using ZEN 2012 software, and all taken images were analyzed with the NIS-Elements software (Nikon, Japan).

### Analysis of blood flow and heart beating in zebrafish

MO-injected zebrafish embryos were treated with 0.2 mM PTU at 12 hpf and recorded using a mono-camera (Nikon, DS-Qi2, Japan) at 48 hpf during 30 s. With the recoded videos, the blood flow and heart beating were analyzed by calculating velocity and beats per minute (bpm), respectively.

### Xenograft in zebrafish

The *cep41* AUG MO-injected or mutant zebrafish embryos were anesthetized with tricaine and transferred onto an agar mold for microinjection after 48 hpf. Several kinds of cancer cells (MDA-MB-435: $3 \times 10^7$, T47D: $2 \times 10^7$, and TC-1: $2.5 \times 10^7$) were labeled with CellTracker CM-DiI (Thermo Fisher Scientific, C7000, USA) and suspended in 20 μl Matrigel solution, and the cells/10 nl were injected near to the perivitelline space where is a superficial location

of the yolk. The zebrafish embryos injected with 10 nl Matrigel solution were applied as a negative control of each xenograft. After the cell transplantation, the embryos were grown in a 28.5°C incubator for 24 h and then were observed and photographed using the confocal microscope (Carl Zeiss, LSM700, Germany). The extra blood vessels generated by transplanted cancer cells were analyzed with the Zeiss ZEN imaging software (Germany).

**Induction of hypoxia in Zebrafish**

To generate hypoxic zebrafish, multigas incubator (Astec, APM-30D, Japan) was used. 1× E3 medium containing 0.017 mg/ml resazurin indicator (TCI, R0203, Japan) was pre-equilibrated in the hypoxia incubator for 24 h. Zebrafish embryos at 28 hpf and then were transferred into the media and incubated for 2 h. After incubation under hypoxic media, the embryos were taken out of the incubator and were immediately transferred into 1.5-ml microtubes. The hypoxic zebrafish embryos were applied for isolation of endothelial cells or immunostaining after washing in 1× PBS twice.

**Isolation of primary endothelial cells and qRT–PCR in Zebrafish**

To isolate primary endothelial cells from *Tg*(*kdrl:EGFP*) zebrafish embryos, about 30–50 embryos were digested with 1 ml of 0.25% trypsin-EDTA (Corning, 25-052-CI, USA) and mechanically dissociated by pipetting. The collected cells were filtered into 5-ml falcon tubes (BD Biosciences, 352235, CA, USA) through 35 μl nylon mesh. Then, DMEM (Corning, 10-013-CVR) supplemented with 10% fetal bovine serum (FBS) (Corning, 35-016-CV) was added to stop further digestion of the cells. The cells were resuspended in DMEM after centrifugation (600 *g* for 2 min at RT) and isolated using a BD FACSAria III cell sorter (BD Biosciences) with the laser set at 488 nm wavelengths. Total RNAs were extracted from the collected endothelial cells with AccuZol™ Total RNA Extraction Solution (Bioneer, K-3090, Korea) according to the manufacturer's instructions. Total RNAs were reverse-transcribed to generate cDNA using the AccuPower® RocketScript™ RT Premix (Bioneer, K-2101, Korea). The mRNA expression levels of the target genes were measured using SYBR green (Bioneer, K-6251, Korea). The cycling parameters were as follows: 95°C for 10 min, followed by 40 cycles of 15 s at 95°C, 30 s at 60°C, and 30 s at 72°C. The mRNA expression level of each gene was normalized with the value of zebrafish *β-actin* mRNA expression. The sequences of primers used for the RT–PCR (Bioneer, Korea) are in Appendix Table S3.

**Statistics**

Statistical analyses were performed and presented using the GraphPad Prism version 8 (GraphPad Software, La Jolla, CA, USA). Values were presented as the mean ± SD or fold change relative to the mean expression in controls. Normally distributed data were subjected to the two-tailed, unpaired Student's *t*-tests with Welch's correction. Two different groups of datasets were subjected to the ANOVA followed by Dunnett's or Tukey's *post hoc* test or to the Brown–Forsythe ANOVA followed by Dunnett's T3 *post hoc* test. Non-normally distributed data were subjected to the Kruskal–Wallis test followed by Dunn's *post hoc* test. A *P*-value of < 0.05 was considered significant for any statistical test used in this study.

**Expanded View** for this article is available online.

## Acknowledgements

This work was supported by the National Research Foundation of Korea (NRF) grant (2017R1E1A2A01076144 and 2018R1A2A3074597 to J.E.L. and 2018R1A4A1024506 to K.W.C., B.-O.C., and J.E.L.) and the BRL grant (NRF 2015041919 to E.-J.C.) funded by the Korean government (MSIP). We thank J.-S. Lee (KRIBB) for *Tg(kdrl:eGFP)* zebrafish; J.-H. Kim and D.-H. Jo (Seoul National University) for advices on angiogenesis analysis; S.-K. Choe (Wonkwang University) for pT7-gRNA vector; H.-S. Park (University of Seoul) for human HIF1α DNA; and H.W.Ko (Dongguk University) for comments on the manuscript.

## Author contributions

SMK, JHK, SYW, SJO, and IYL conducted experiments and analyzed data. Y-KB, KWC, B-OC, E-JC, and BP advised on experimental designs and commented on the manuscript. JEL designed overall experiments with substantial contribution from SMK, JHK, and SYW and wrote the manuscript.

## Conflict of interest

The authors declare that they have no conflict of interest.

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
