## [Review Process File · EMBO Reports]

CEP41-mediated ciliary tubulin glutamylation drives angiogenesis through AURKA-dependent deciliation

Soo Mi Ki, Ji Hyun Kim, So Yeon Won, Shin Ji Oh, In Young Lee, Young-Ki Bae, Ki Wha Chung, Byung-Ok Choi, Boyoun Park, Eui-Ju Choi, & Ji Eun Lee

Review timeline:

Submission date:	15 April 2019
Editorial Decision:	24 May 2019
Revision received:	1 October 2019
Editorial Decision:	13 November 2019
Revision received:	18 November 2019
Accepted:	29 November 2019

Editor: Deniz Senyilmaz-Tiebe

Transaction Report:

1st Editorial Decision

24 May 2019

Thank you for submitting your manuscript for consideration by EMBO Reports. Three referees agreed to review your manuscript. So far, we have received two referee reports that are copied below. Given that both referees are in fair agreement that you should be given a chance to revise the manuscript, I would like to ask you to begin revising your study along the lines suggested by the referees.

Please note that this is a preliminary decision made in the interest of time, and that it is subject to change should the third referee offer very strong and convincing reasons for this. As soon as we will receive the final report on your manuscript, we will forward it to you as well.

As you can see, both referees express interest in the proposed role of CEP41 in regulation of tubulin glutamylation in the cilia and angiogenesis under shear stress. However, they also raise concerns that need to be addressed in full before we can consider publication of the manuscript here.

Given these constructive comments, I would like to invite you to revise your manuscript with the understanding that the referee must be fully addressed and their suggestions taken on board. Please address all referee concerns in a complete point-by-point response. Acceptance of the manuscript will depend on a positive outcome of a second round of review. It is EMBO Reports policy to allow a single round of revision only and acceptance or rejection of the manuscript will therefore depend on the completeness of your responses included in the next, final version of the manuscript.

REFeree REPORTS

Referee #1:

In the paper by Soo Mi Ki et al, the authors describe a novel mechanism important for angiogenesis.

The authors identified that the cilia related protein cep41 controls tubulin glutamylation of the cilium of endothelial cells, which regulates the angiogenic response of endothelial cells upon shear stress. The authors identified that the cep41-aurora kinase-hif pathway is important for angiogenesis not only during development, but also during hypoxia and tumorigenesis. The authors used cell culture and zebrafish as model system with a combination of siRNA, MO and CrispR mutant to support their claims. In general, the manuscript is well written with the questions put in a broad context and understandable for non-experts. The findings are interesting and bridge multiple fields. Therefore, I recommend this manuscript for publication. However, I have listed a set of comments below that should be addressed before the manuscript is accepted for publication.

Major comments

- 1) The authors perform many combinations of knock-down and overexpression which are very informative. However, I am surprised that in many instance the authors do not provide results obtained upon the single knock-down or single overexpression. This include the effect of CCP5 knock-down or AURKA overexpression in cell culture. See also comments #2 and #3.
- 2) In Fig S5, the authors show that CCP5 interfere with angiogenesis in zebrafish. Is this the case also in cell culture? It is important that the authors report the effect of CCP5 knock-down alone on ciliogenesis and glutamylation of tubulin. It is not very clear how it is possible that both single knock-down have angiogenesis phenotypes and that upon double knock-down (cep41, ccp5) the angiogenesis phenotype is rescued. The authors can better discuss these findings.
- 3) In Fig 6I-M, it is important to report the effects of overexpression of AURKA on the angiogenesis phenotype in control siRNA samples. Does it enhance tube formation and wound closure?
- 4) In many quantifications (e.g 5G-J, 6A-B, 7B-C,F-G, 8J-K), the authors compare the values obtained from siRNA samples after treatment with the values obtained from the control siRNA before treatment. Would it not be more appropriate to compare the value before and the value after treatment for each respective siRNA/MO?
- 5) In FigS7 and S9, the authors show that shear stress and hypoxia induce high levels of glutamylation in cilia of HUVEC, and that the cilia with higher glutamylated tubulin are longer than the other cilia. However, in Fig5J, the authors show that the remaining cilia upon serum starvation have low levels of glutamylated tubulin. These results appear somehow contradictory. I suggest that the authors discuss these findings better.
- 6) In Fig8E, expression of VEGFA and VEGFR2 is only mildly rescued in siCEP41 kd cells upon re-expression of CEP41, when compared to rescue with HIF1alpha or other factors. This poor rescue has serious consequences for other experiments performed in the manuscript. For instance, how would tube formation or wound healing defects be rescued by expression of CEP41 alone in CEP41 knocked-down cells? Is it possible that HIF1a and AURKA have an effect beyond CEP41 and thus have a better rescuing ability? The authors need to provide the adequate controls and better discuss these results.

Other comments

- 7) In FigS2, the authors show that cep41 MO injected zebrafish have a deformed morphology. How does the cep41 homozygous mutant look like? Do they present the typical curved body axis of cilia mutant?
- 8) In page 6, the authors refer to the caudal plexus. It may be good to indicate the location of the caudal plexus on the fig S2 to help orient non-experts.
- 9) In Fig 3, the authors refer to the relative number of cilia per field. It may be more appropriate to report the absolute number of cilia instead of the relative number.
- 10) In all figures, avoid using green and red. Particularly for figure 6G and figure 7D, it is very difficult to see the red dots. I recommend for those particular figures to split the colors into different panels.
- 11) In FigS9, quantification of cilia resorption upon hypoxia should be included similarly to the other treatments.
- 12) In Fig 6K-M, AURKA overexpression is sufficient to rescue the angiogenesis phenotype but not in Fig7J. These results are barely touched upon and may be better discussed in the context of the Fig8L.

13) In Fig8C, why is there a strong signal in the IgG IP for HIF1a and not in the HIF1a lanes?

Referee #3:

Overall, this manuscript presents an interesting hypothesis linking shear stress to CEP41-controlled tubulin glutamylation and downstream events including AURKA-activation and upregulation of angiogenic factors VEGFA and VEGFR2. The data presented linking CEP41 to vasculogenesis via regulation of glutamylation of cilia, and in particular the rescue via knockdown of the CCP5 deglutamylase are exciting. However, the manuscript has major concerns outlined below:

1. There is a failure to directly link CEP41 knockdown to an altered response to shear stress. The data shown in Figure6C needs better quantification (are the white numbers the quantification normalized to control non-shear stress?), the image shown is overexposed: this data is absolutely central to linking shear stress-CEP41-pAURKA. At a minimum, the data needs improved quantification, evidence of statistical significance over more than a single experiment and non-shear stress ctrl data for the CEP41 siRNA.
 2. In general, the data on cilia resorption in response to shear stress and glutamylation state is suggestive. However, the presentation is unclear. For example, 5 H,I,J are incredibly confusing. I think the authors are trying to show that knockdown of CCP5 allows cilia resorption in the face of CEP41 KD by permitting deglutamylation. Does this mean the baseline level of glutamylation is higher in the siCEP41/siCCP5 cells than the siCEP41 cells? The figure might be made more interpretable by including the baseline (serum-starved) level of Arl13b+ cells and GT335(-) cells for all three conditions (ctrl, siCEP41 and siCEP41/siCCP5), including fewer time points, and separating the groupings in the bar graphs by time points.
 3. Throughout the manuscript the authors seem to equate the ciliary response to serum repletion to the response to shear stress. Although this is possible, it is not definitive and needs to be better described.
 4. The work attempting to link CEP41 and glutamylation to downstream effectors of angiogenesis is difficult to interpret. The evidence that there is an effect on VEGF regulation is ok, however, in Fig.6D why do the authors not show the serum-starved baseline for CEP41 siRNA? It looks like pAURKA is also increased in CEP41 siRNA cells similar to control in response to feeding; why not show baseline for both? Does this mean that pAURKA activation is CEP41 dependent in response to shear stress, but not serum refeeding?
 5. Zebrafish data shown in Fig6G is unclear. The pAURKA signal is difficult to see on the merged image, the figure legend does not indicate the color scheme (red for pAURKA?). In addition, the most visible red signal is in the CtrlMO image in an area devoid of cells, making it difficult to interpret the significance of this finding.
 6. Figure 6K looks as if AURKA rescued siCEP41 as effectively as activated AURKA T288D; what does this mean?
 7. The hypoxia data does not mechanistically link to the shear stress data, and the whole section feels like it was added on to the main concept of the manuscript. Hypoxia has many effects that can confound the findings. For example, what happens to flow and the resultant shear stress in zebrafish exposed to hypoxia?
 8. The epistasis between CEP41, HIF1a and VEGF presented in Fig 8 is unclear.
 9. The data on the relationship between sprouting angiogenesis in tumors and CEP41 is very preliminary, and does not add to the current manuscript.
 10. Throughout the manuscript, the authors fail to show control data for cells and embryos with CEP41 knockdown. This makes it very difficult to interpret many of the bar graphs, as the reader is asked to compare, for example, serum-starved control cells to serum-fed CEP41 knockdown cells.
- Minor points:
11. Fig 2B-The diameter of ISV lumen looks very comparable between CEP41 KD and CTRL, however, the architecture is strikingly abnormal. Is this a representative image?
 12. Fig. 3: what about other parameters of cilia structure such as length?
Note that the decrease in glutamylation in response to CEP41 knockdown appears to be restricted to the distal cilium while the proximal cilium seems to retain GT335. Does this matter? Is this region the transition zone? Note that this GT335 signal appears to be lost in the in-vivo images shown in 3C.
 13. Tubulin glutamylation in response to hypoxia: the data in Fig. S9 looks more like decrease in Arl13b, than increase in GT335; not sure whether this figure actually shows increase in tubulin

glutamylated in response to hypoxia

14. There are many grammatical errors throughout, and they are especially apparent in the discussion section.

In summary, this manuscript presents an exciting hypothesis and would be greatly improved by focusing on the link between shear stress, CEP41, cilia glutamylation and vasculogenesis. The single most important improvement would be adequately buttressing the data linking shear stress to CEP41, cilia glutamylation and vasculogenesis. The data on hypoxia and tumorigenesis is incomplete, and confuses the main point the authors are making. In addition, the presentation of immunofluorescence images is poor, and makes it difficult to interpret some of the data. Finally, the appropriate controls are not routinely presented as part of the bar graphs, again making interpretation of the data very difficult.

1st Revision - authors' response

1 October 2019

Referee #1:

In the paper by Soo Mi Ki et al, the authors describe a novel mechanism important for angiogenesis. The authors identified that the cilia related protein cep41 controls tubulin glutamylation of the cilium of endothelial cells, which regulates the angiogenic response of endothelial cells upon shear stress. The authors identified that the cep41-aurora kinase-hif pathway is important for angiogenesis not only during development, but also during hypoxia and tumorigenesis. The authors used cell culture and zebrafish as model system with a combination of siRNA, MO and CrispR mutant to support their claims. In general, the manuscript is well written with the questions put in a broad context and understandable for non-experts. The findings are interesting and bridge multiple fields. Therefore, I recommend this manuscript for publication. However, I have listed a set of comments below that should be addressed before the manuscript is accepted for publication.

- We deeply appreciate the referee's positive feedback on our work.

Major comments

1) The authors perform many combinations of knock-down and overexpression which are very informative. However, I am surprised that in many instance the authors do not provide results obtained upon the single knock-down or single overexpression. This include the effect of CCP5 knock-down or AURKA overexpression in cell culture. See also comments #2 and #3.

- According to the referee's comments, in the revised manuscript, we have included the data on the effects of *CCP5* knockdown and *AURKA* overexpression in cultured cells. Please refer to the responses to comments #2 and #3 for the interpretation of each experimental result.

2) In Fig S5, the authors show that CCP5 interfere with angiogenesis in zebrafish. Is this the case also in cell culture? It is important that the authors report the effect of CCP5 knock-down alone on ciliogenesis and glutamylation of tubulin. It is not very clear how it is possible that both single knock-down have angiogenesis phenotypes and that upon double knock-down (cep41, ccp5) the angiogenesis phenotype is rescued. The authors can better discuss these findings.

- We appreciate the referee's critical comments on the rescue data with *CCP5* interference. According to the referee's suggestions, we have analyzed the effect of *CCP5* single knockdown on cilia assembly/disassembly/glutamylation as well as on angiogenesis. Consistent with the data from zebrafish, the single *CCP5* depletion also caused angiogenic defects, including impairment of wound closure and tube formation in HUVECs (Fig 4A-E). Regarding to its effect on the cilia, we found while *CEP41* depletion reduces the levels of ciliary glutamylation, *CCP5* depletion increases the glutamylation levels in the cilia (Fig 4I). Moreover, we found that the altered ciliary glutamylation levels are restored by the double knockdown of *CEP41* and *CCP5* (Fig 4I). As expected, *CCP5* depletion had little effect on cilia assembly (Fig 4H and 5I, J). Intriguingly, as shown in *CEP41* depletion, *CCP5* depletion resulted in suppression of cilia disassembly (Fig 5J). In addition, we found that the inhibited cilia disassembly is restored by double knockdown of *CEP41* and *CCP5* (Fig 5J). Notably, we found that the opposing effects between *CEP41* and *CCP5* on

ciliary glutamylation level are neutralized by simultaneous depletion of both even under deciliation condition (Fig 5K). Thus, we suggest that balance of CEP41 and CCP5 in the cilia is critical for appropriate levels of tubulin glutamylation, and the balanced ciliary tubulin glutamylation controls cilia disassembly to regulate angiogenesis. A more detailed discussion about these results has been described in the sentence “Of note, *CCP5* depletion...” on page 18 of the main text. Please also refer to the response to comment #2 of the referee #3.

3) In Fig 6I-M, it is important to report the effects of overexpression of AURKA on the angiogenesis phenotype in control siRNA samples. Does it enhance tube formation and wound closure?

- To address the referee's question, we have investigated the effects of AURKA overexpression on the angiogenesis in control cells using tube formation analysis, and compared the results between static and shear stress conditions. Under a static state, exogenous AURKA hardly affected tubulogenesis in control cells (Fig 6K and L); in contrast, under a shear stress state, it enhanced tube formation as much as control cells overexpressing the active form of AURKA (T288D) (Fig 6M and N). The referee #3 also raised the relevant comments. Please refer to the response to comment #6 of the referee #3 for an overall interpretation of these experimental results.

4) In many quantifications (e.g 5G-J, 6A-B, 7B-C,F-G, 8J-K), the authors compare the values obtained from siRNA samples after treatment with the values obtained from the control siRNA before treatment. Would it not be more appropriate to compare the value before and the value after treatment for each respective siRNA/MO?

- We apologize for inappropriate quantitative analyses in the previous manuscript. According to the referee's suggestions, the resulting data before and after treatments for each siRNA/MO samples were properly compared for statistical analyses. Finally, we have included new quantification data in Fig 5G-K, 6A,B,E,F,J, and K-N, 7B,C,F,G, and I-J, 8 E,F,H-K of the revised manuscript.

5) In FigS7 and S9, the authors show that shear stress and hypoxia induce high levels of glutamylation in cilia of HUVEC, and that the cilia with higher glutamylated tubulin are longer than the other cilia. However, in Fig5J, the authors show that the remaining cilia upon serum starvation have low levels of glutamylated tubulin. These results appear somehow contradictory. I suggest that the authors discuss these findings better.

- We agree that the interpretation of the data presented in previous Fig S7, S9, and Fig 5J is complicated. Considering all relevant comments (#11 of the referee #1 and #2 of the referee #3), we re-designed experiments to examine the ciliary glutamylation levels. Indeed, we have found that longer cilia of the GT335+/ARL13b+ cells were due to shortened ARL13b-labeled cilia than changes in GT335-labeled cilia length. Thus, by double immunostaining with ARL13b and GT335 antibodies, we have measured the signal intensities for each antibody in the double-labeled cilia. We found while ciliary ARL13b intensity is not affected by the following treatments (serum, shear stress, and hypoxia), ciliary GT335 intensity is increased by these treatments. In other words, all of treatments consistently caused higher glutamylation levels in the cilia. We have included these data in Fig EV3, EV4, and Fig 5 H, K of the revised manuscript.

6) In Fig8E, expression of VEGFA and VEGFR2 is only mildly rescued in siCEP41 kd cells upon re-expression of CEP41, when compared to rescue with HIF1alpha or other factors. This poor rescue has serious consequences for other experiments performed in the manuscript. For instance, how would tube formation or wound healing defects be rescued by expression of CEP41 alone in CEP41 knocked-down cells? Is it possible that HIF1a and AURKA have an effect beyond CEP41 and thus have a better rescuing ability? The authors need to provide the adequate controls and better discuss these results.

- To improve the quality of data, qRT-PCRs have been repeated with the *CEP41* knockdown cells transfected with different concentrations of CEP41 expression vectors. We found that expression of both *VEGFA* and *VEGFR2* mRNAs depends on the amount of exogenous CEP41 expression (Please refer to the data below here). Accordingly, we have replaced the Fig 8E and F with fully rescued data in the revised manuscript.

A, B The mRNA levels of *VEGFA* (**A**) and *VEGFR2* (**B**) quantified by qRT-PCR in control or *CEP41*-knockdown HUVECs overexpressing *CEP41* under normoxia or hypoxia conditions. The expression of *GAPDH* was quantified for the normalization of the qRT-PCR results. Data are mean \pm SD of three independent experiments. Statistical significance was determined with the Brown-Forsythe ANOVA followed by Dunnett's T3 *post hoc* test ($^{***}P < 0.001$, ns: non-significant).

- Under hypoxia, overexpression of *AURKA* failed to restore the *CEP41* knockdown-derived tube formation defects, whereas overexpression of active form of *AURKA* (T288D) partially restored the defects (Fig 7H-J). Moreover, we found that overexpression of *HIF1a* partially restored the tubulogenesis defects but simultaneous overexpression with *CEP41* fully restored the defects (Fig 7H-J). Thus, these results indicated that activation of *AURKA* and *HIF1a* is essential for *CEP41*-mediated angiogenesis under hypoxia. By further studies, the partial rescue by *HIF1a* overexpression was explained by the finding that *CEP41* activates *AURKA* by preventing UPS-dependent degradation of *HIF1a* (Fig 8D, G).

Other comments

7) In FigS2, the authors show that *cep41* MO injected zebrafish have a deformed morphology. How does the *cep41* homozygous mutant look like? Do they present the typical curved body axis of cilia mutant?

- We apologize for missing the data about morphological phenotypes of *cep41* homozygous mutants in the previous manuscript. In the revised manuscript, we have included the *cep41* mutant data showing ciliary phenotypes, such as hydrocephalus, hart edema, and curved body axis, which are similar to those of *cep41* morphants in Appendix Fig S2I.

8) In page 6, the authors refer to the caudal plexus. It may be good to indicate the location of the caudal plexus on the fig S2 to help orient non-experts.

- Caudal plexus meant caudal vein plexus. We apologize for confusion between these terminologies. We have corrected the term in the main text and indicated the location on the revised Appendix Fig S2J.

9) In Fig 3, the authors refer to the relative number of cilia per field. It may be more appropriate to report the absolute number of cilia instead of the relative number.

- According to the referee's suggestion, we have changed the relative number of cilia to the absolute number of cilia in the revised Fig 3D and Fig EV1F.

10) In all figures, avoid using green and red. Particularly for figure 6G and figure 7D, it is very difficult to see the red dots. I recommend for those particular figures to split the colors into different panels.

- To avoid using green and red colors in the same panel, we have merged the immunostained-phosphorylated *AURKA* (red) with the nucleus of endothelial cell (blue). Furthermore, we have generated 3D images with the original immunostaining data and shown clearly the red dots

(phospho-AURKA) in high quality images. The data in Fig 6G and Fig 7D have been replaced with the 3D images.

11) In FigS9, quantification of cilia resorption upon hypoxia should be included similarly to the other treatments.

- According to the referee's suggestion, the ciliary resorption upon hypoxia was quantified by counting ARL13b-positive cells (ciliated cells) that were also applied for quantification of deciliation upon shear stress (Fig 5G) and serum retrieval (Fig 5J). We have included the new data in the revised Fig EV4B.

12) In Fig 6K-M, AURKA overexpression is sufficient to rescue the angiogenesis phenotype but not in Fig7J. These results are barely touched upon and may be better discussed in the context of the Fig8L.

- By further experiments including appropriate controls, we have better understood the role of CEP41 in activation of AURKA in shear stress and hypoxia conditions. Consistent with the previous results, AURKA overexpression was sufficient to rescue the *CEP41* depletion-induced angiogenic defects in shear stress condition (Fig 6M, N), but not in hypoxia condition (Fig 7H-J). As briefly described on response to comment #6, we have addressed that CEP41 is required for HIF1a stabilization to activate AURKA under hypoxia. Thus, non-recovery of angiogenic defects by AURKA overexpression is likely due to the hypoxia-induced HIF1a being prone to degradation in the absence of *CEP41* and thereby unable to activate exogenous AURKA in hypoxia. However, in shear stress, the effects of AURKA overexpression recovering angiogenic defects in the absence of *CEP41* were unexpected. Because we have found that *CEP41* depletion inhibited the activation of AURKA and the increase of *VEGFA/R2* mRNA expressions under shear stress (Fig 6A-C). Taken together, this data imply that shear stress may activate exogenous AURKA independently of CEP41. We have described a more detailed discussion of these results in the sentence "Moreover, we found that only under shear stress,..." on page 20 of the revised manuscript.

- We agree that the diagram presented in the previous Fig 8L is somewhat beyond our findings. Based on the new data in revised Fig 7 and 8, the schematic model has been substituted by a diagram that suggests the role of CEP41 in preventing HIF1a degradation in the revised manuscript.

13) In Fig8C, why is there a strong signal in the IgG IP for HIF1a and not in the HIF1a lanes?

- To eliminate the non-specific signals shown in the IgG IP for HIF1a, we repeated the co-IP experiments under optimized conditions: the IP washing buffer was changed from PBS w/ 0.02% Tween 20 to PBS w/ 0.5% Tween 20, and washing time was increased from 3X10 min to 3 X15 min. The previous data in Fig 8C have been replaced with new blot images.

Referee #3:

Overall, this manuscript presents an interesting hypothesis linking shear stress to CEP41-controlled tubulin glutamylation and downstream events including AURKA-activation and upregulation of angiogenic factors VEGFA and VEGFR2. The data presented linking CEP41 to vasculogenesis via regulation of glutamylation of cilia, and in particular the rescue via knockdown of the CCP5 deglutamylase are exciting. However, the manuscript has major concerns outlined below:

- We deeply appreciate the referee's positive feedback on our work.

1. There is a failure to directly link CEP41 knockdown to an altered response to shear stress. The data shown in Figure6C needs better quantification (are the white numbers the quantification normalized to control non-shear stress?), the image shown is overexposed: this data is absolutely central to linking shear stress-CEP41-pAURKA. At a minimum, the data needs improved quantification, evidence of statistical significance over more than a single experiment and non-shear stress ctrl data for the CEP41 siRNA.

- According to the referee's suggestion, we have performed more than three western blottings for pAURKA in control and *CEP41* knockdown cells under static and shear stress conditions. The results showed repeatedly that pAURKA levels increase in control cells responding to shear stress compared to the static control, whereas *CEP41* depletion interferes with the shear stress-induced increase of pAURKA levels. We have included the new immunoblot image data and the quantification data in the revised Fig 6C and the Appendix Fig S5A, respectively.

2. In general, the data on cilia resorption in response to shear stress and glutamylation state is suggestive. However, the presentation is unclear. For example, 5 H,I,J are incredibly confusing. I think the authors are trying to show that knockdown of CCP5 allows cilia resorption in the face of CEP41 KD by permitting deglutamylation. Does this mean the baseline level of glutamylation is higher in the siCEP41/siCCP5 cells than the siCEP41 cells? The figure might be made more interpretable by including the baseline (serum-starved) level of ARL13b+ cells and GT335(-) cells for all three conditions (ctrl, siCEP41 and siCEP41/siCCP5), including fewer time points, and separating the groupings in the bar graphs by time points.

- We appreciate the referee's critical comments on the data in Fig 5H-J of the previous manuscript. According to the referee's suggestions, we have measured ciliary disassembly and glutamylation levels in the siCtrl, siCEP41, siCCP5, and siCEP41/CCP5-transfected RPE1 cells in serum - and serum + conditions. In serum starvation, the depletion of CEP41 and CCP5 hardly affected cilia assembly; in contrast, in serum retrieval, both CEP41 and CCP5 depletion prevented cilia disassembly. Of note, the simultaneous depletion of CEP41 and CCP5 rescued the hampered deciliation in serum + condition (Fig 5I and J).

- We agree that the interpretation of the data presented in previous Fig 5H-J is complicated. As described in response to comment #5 of the referee #1, we have examined the ciliary glutamylation level by analyzing the GT335 signal intensity normalized to the ARL13b intensity. Control ciliary glutamylation levels were increased by serum addition; whereas, the glutamylation levels of CEP41-depleted cilia were decreased under serum starvation and the reduced levels were maintained even under serum addition. Remarkably, CCP5 depletion increased ciliary glutamylation levels in serum starvation and further increased the levels after serum addition. These altered ciliary glutamylation levels were restored to normal levels by double knockdown of CEP41 and CCP5 in each serum condition (Fig 5K). Overall, this data suggested that a careful balance of CEP41 and CCP5 controls ciliary glutamylation levels and the proper levels is important for control of cilia disassembly. A more detailed explanation and discussion have been described in the sentence "To understand how CEP41 and CCP5 work together..." on page 10 and in the sentence "Of note, CCP5..." on page 18 of the main text.

- Previously, we counted the number of cells with GT335(-)/ARL13B(+) cilia, along with hypothesis that inhibition of ciliary resorption (ARL13b+) might simply be due to a deficiency of ciliary glutamylation (GT335-) in serum + condition. However, the results of GT335 intensity and deciliation shown in CCP5 depletion have suggested that glutamylation level, rather than glutamylation itself, is the major determinant for cilia disassembly. Indeed, the cells with GT335(-)/ARL13b(+) cilia were few in CCP5-depleted cells under serum + condition (i.e., the retained cilia in CCP5-depleted cells were still glutamylated to high levels). Accordingly, we agree that the counting data of GT335(-)/ARL13b(+) ciliated cells is rather complicated to interpret, especially when compared to the results of serum - condition. Finally, we have decided to replace the counting data with intensity data in the revised manuscript (Fig 5K). For referee's estimation, we present the data, including the CCP5 depletion, below here.

Quantification of GT335-negative cells within ARL13b-positive cells under either serum starvation (0 h) or serum retrieval (8 and 18 h) conditions are the results of three independent experiments with ≥ 200 cells per condition (mean \pm SD). *** $P < 0.001$, ns: non-significant (Two-way ANOVA with Tukey's *post hoc* test).

3. Throughout the manuscript the authors seem to equate the ciliary response to serum repletion to the response to shear stress. Although this is possible, it is not definitive and needs to be better described.

- In our studies, we have described the serum retrieval and shear stress as stimulators of cilia disassembly in RPE1 cells and HUVECs, respectively. We generally agree that common features appeared in different conditions are not always due to the function of equivalent mechanisms. Thus, in the revised manuscript, we have carefully described whether the ciliary response is due to serum retrieval or shear stress.

4. The work attempting to link CEP41 and glutamylation to downstream effectors of angiogenesis is difficult to interpret. The evidence that there is an effect on VEGF regulation is ok, however, in Fig.6D why do the authors not show the serum-starved baseline for CEP41 siRNA? It looks like pAURKA is also increased in CEP41siRNA cells similar to control in response to feeding; why not show baseline for both? Does this mean that pAURKA activation is CEP41 dependent in response to shear stress, but not serum refeeding?

- According to the referee's comments, we have performed more than three western blot assays for pAURKA and measured the pAURKA levels of the control and *CEP41*-depleted cells before and after each treatment. We found that basal levels of pAURKA are not affected by *CEP41* depletion in both serum starvation and static states (i.e., before treatment); In contrast, in both serum addition and shear stress conditions (i.e., after treatment), *CEP41* depletion interferes with an increase of pAURKA levels (Fig 6C and D). Thus, these results suggest that activation of AURKA is CEP41 dependent both in shear stress and serum retrieval. The new immunoblot image data and quantification data have been included in the revised manuscript (Fig. 6C and D; Appendix Fig S5A and B).

5. Zebrafish data shown in Fig6G is unclear. The pAURKA signal is difficult to see on the merged image, the figure legend does not indicate the color scheme (red for pAURKA?). In addition, the most visible red signal is in the CtrlMO image in an area devoid of cells, making it difficult to interpret the significance of this finding.

- We apologize for presenting unoptimized image data and for the insufficient description of the data. The referee #1 also commented on this image data, so we addressed it according to referee #1's suggestion. We have merged the immunostained-pAURKA (red) with the nucleus of endothelial cell (blue) to avoid using green and red colors in the same panel. Moreover, we have generated 3D images of the immunostaining data to clearly show the red dots (phospho-AURKA) in high quality images. The previous images have been substituted with new 3D images in the revised Fig 6G and Fig 7D.

6. Figure 6K looks as if AURKA rescued siCEP41 as effectively as activated AURKA T288D; what does this mean?

- To better address the effective rescue of tubulogenesis by exogenous AURKA in *siCEP41*-transfected cells under shear stress, we have examined the effects of exogenous AURKA on tubulogenesis of control and *CEP41*-depleted cells under static states. We found that in static conditions, unlike the results of shear stress conditions, the expression of exogenous AURKA did not affect the tube formation in both control cells and *CEP41*-depleted cells. Moreover, repeated experiments led to the conclusion that tubulogenesis defects in *CEP41*-depleted cells are not restored by exogenous AURKA as much as normal tubulogenesis under shear stress (Fig 6M and N). In addition to this new data on AURKA overexpression, we found that in both static and shear stress conditions, the expression of exogenous AURKA (T288D) enhances the tube formation in control cells and restores the defective tubulogenesis as much as normal tubulogenesis in *CEP41*-depleted cells (Fig 6K-N). In this revision, we have manually validated the quantification data for tube numbers and lengths obtained through automatic analysis using ImageJ software. Thus, the results of AURKA overexpression different from the previous manuscript may be due to false positive data derived from automated analysis.

- We don't understand why, but the data on AURKA overexpression led us to speculate that shear stress may affect activation of exogenously introduced AURKA independent of CEP41. Please refer to the detailed discussion described in the sentence "Moreover, we found that only under shear stress,..." on page 20 in the revised manuscript. Although the mechanism by which shear stress activates exogenous AURKA without CEP41 needs to be identified in future studies, we have found that in the absence of *CEP41*, AURKA is not activated and mRNA expression of *VEGFA/R2* is not increased by shear stress (Fig 6A-C). Therefore, our data indicate that CEP41 is required for AURKA activation to induce angiogenesis under shear stress.

7. The hypoxia data does not mechanistically link to the shear stress data, and the whole section feels like it was added on to the main concept of the manuscript. Hypoxia has many effects that can confound the findings. For example, what happens to flow and the resultant shear stress in zebrafish exposed to hypoxia?

- Here, we have claimed that regulation of *VEGFA/R2* expression via AURKA activation is a key molecular mechanism of CEP41-mediated angiogenesis functioning in shear stress and hypoxia. Although we generally agree that hypoxia is involved in many ways in regulating angiogenesis, we have demonstrated that CEP41 is required for prevention of UPS-dependent degradation of HIF1 α and that this is essential to regulate the AURKA-VEGF molecular axis.

- To address the referee's concern, we have investigated the effect of hypoxia on blood vessel formation in each zebrafish responding to low or high shear stress. As shown below here, hypoxia exposure markedly enhanced angiogenesis of control zebrafish under low shear stress; in contrast, it had little effect on zebrafish angiogenesis under high shear stress. This data indicates that once angiogenesis is induced by shear stress, the angiogenesis is hardly affected by hypoxia, suggesting that there is a common mechanism activated by shear stress and hypoxia in regulation of zebrafish angiogenesis. Indeed, by rough comparison of the data between normoxia and hypoxia, we found hypoxia does not significantly activate AURKA in control zebrafish under high shear stress (Fig 6H vs Fig 7E). Moreover, we found that under low shear stress, hypoxia exposure does not enhance angiogenesis in *cep41*-deficient zebrafish, which is defective in shear stress response. Notably, under high shear stress, we found that the AURKA activation and the *VEGFA/R2* expression are suppressed in *cep41*-deficient zebrafish exposed to hypoxia (Fig 7D-G). Thus, our data suggest that CEP41 regulates angiogenesis via the AURKA-VEGF pathway in both shear stress and hypoxia conditions, although the mediators of the CEP41-AURKA-VEGF pathway may differ between the two conditions.

A. The graph reveals shear stress values in control and *cep41*-MOs injected zebrafish at the indicated developmental time points. **B** and **C.** The graphs display quantified data of ISV length in the indicated MO-injected zebrafish under normoxia (NOR) or hypoxia (HYP) at 24 hpf (low shear stress state) or 30 hpf (high shear stress state). The data are shown as the mean \pm SD; *** P < 0.001, ns: non-significant. Over 20 zebrafish embryos were analyzed for the experiments. (Kruskal-Wallis test with Dunn's *post hoc* test).

8. The epistasis between CEP41, HIF1 α and VEGF presented in Fig 8 is unclear.

- We agree that the diagram presented in the previous Fig 8L is somewhat beyond our findings, although we proposed the model based on the results of previous Fig 7H-J and 8D. Instead, based on the new data in revised Fig 7 and 8, the schematic model has been substituted by a diagram that suggests the role of CEP41 in preventing HIF1 α degradation in the revised manuscript.

9. The data on the relationship between sprouting angiogenesis in tumors and CEP41 is very preliminary, and does not add to the current manuscript.

- We have performed this experiment to investigate if CEP41 is also involved in angiogenesis under tumor-induced hypoxia. We generally agree that this data is slightly beyond the scope of main finding in our studies, so we have moved this data to Appendix Fig S9 in the revised manuscript.

10. Throughout the manuscript, the authors fail to show control data for cells and embryos with CEP41 knockdown. This makes it very difficult to interpret many of the bar graphs, as

the reader is asked to compare, for example, serum-starved control cells to serum-fed CEP41 knockdown cells.

- We apologize for the difficulty in interpreting the results of missing some essential control data in the previous manuscript. According to the referee's suggestions, we have included proper controls in most experiments and performed statistical analyses with appropriate comparisons in the revised manuscript.

Minor points:

11. Fig 2B-The diameter of ISV lumen looks very comparable between CEP41 KD and CTRL, however, the architecture is strikingly abnormal. Is this a representative image?

- While it is consistent that ISVs in both *cep41* KD and KO zebrafish are markedly impaired, not all ISVs of *cep41*-deficient zebrafish are defective. That is why some ISVs in the *cep41* KD/KO zebrafish images show mild phenotypes. Nevertheless, we agree that presentation of better representative images, consistent with the quantification data of the ISV lumen size, is required. Accordingly, the images of *cep41* morphants and *cep41* mutant zebrafish have been substituted in the revised manuscript (Fig 2A). In addition, we have marked the representative ISVs with dotted rectangles on the images to clarify the difference in sizes of the ISV lumens

12. Fig. 3: what about other parameters of cilia structure such as length?

Note that the decrease in glutamylation in response to CEP41 knockdown appears to be restricted to the distal cilium while the proximal cilium seems to retain GT335. Does this matter? Is this region the transition zone? Note that this GT335 signal appears to be lost in the in-vivo images shown in 3C.

- The majority of *CEP41*-depleted cells show nearly complete loss of GT335-labeled cilia, but a few of KD cells have glutamylated tubulin in regions that appear to be proximal cilia. Thus, the length of glutamylated cilia was measured only in the cells with GT335 signal at the regions look like proximal axoneme. The data revealed that length of both whole cilia (ARL13b-marked) and glutamylated cilia (GT335-marked) shortens in the *CEP41* KD cells. We have included this data in the revised manuscript (Fig 4J).

- We assume that the referee might regard the GT335-labeled centrosomes as the GT335-labeled proximal cilia in the *CEP41*-depleted cells. The centrosomal structures are indistinguishable in some of cell images, since they look more like GT335-stained lines than double dots. To determine where the GT335 signal comes from in the *CEP41*-deficient cells, we have performed double-immunostaining with GT335 and a marker of transition zone (CEP290) or a marker of centrosome (g-Tub). As shown below here, we found that in the *CEP41* KD cells, the GT335 signals largely overlap the g-Tub signals in the centrosomes; in contrast, the GT335 signals barely overlap with CEP290 signals that appear to be in the transition zones. Thus, we think that the GT335 signals retained in *CEP41*-depleted cells are not limited to the transition zone but rather derived from the centrosome.

A and B. HUVECs were transfected with control or *CEP41* siRNAs and then immunostained with anti-CEP290 and GT335 (**A**) or with anti- γ -Tub and GT335 (**B**) antibodies. Asterisks and arrows indicate transition zones and centrosomes, respectively. Scale bars, 20 μ m.

- As the referee guesses, the analysis of the cilia length is quite difficult *in vivo*. In particular, it is almost impossible to distinguish cilia and centrosomes by GT335 staining within tissues. Thus, we have analyzed 3D images converted from the original staining image files and found that GT335 signals not observed in the planar images are visible in the side view of the 3D images (Please see the data **B** and **C** below here). Accordingly, we think that in analysis of the planar images, the GT335 signal on the protruded cilia of the wild type zebrafish was clearly detected, while the GT335 signal on the relatively flat centrosomes or proximal axoneme of the *cep41*-morphants and mutants was not detected.

13. Tubulin glutamylation in response to hypoxia: the data in Fig. S9 looks more like decrease in Arl13b, than increase in GT335; not sure whether this figure actually shows increase in tubulin glutamylation in response to hypoxia.

- Referee #1 also raised similar comments on this figure. Please refer to our response to comment #5 of the referee #1.

- In brief, we re-designed the experiments to investigate the ciliary glutamylation levels, since we found that in hypoxia-responded cells, the ARL13b-labeled cilia shorten, but the GT335-labeled cilia retain their length. Thus, the levels of ciliary glutamylation were determined by analyzing the GT335-signal intensity rather than the length of the cilia.

14. There are many grammatical errors throughout, and they are especially apparent in the discussion section.

- This manuscript has been revised again through the American English Editing Service. We think the grammatical errors in the previous version have been corrected in this version.

In summary, this manuscript presents an exciting hypothesis and would be greatly improved by focusing on the link between shear stress, CEP41, cilia glutamylation and vasculogenesis. The single most important improvement would be adequately buttressing the data linking shear stress to CEP41, cilia glutamylation and vasculogenesis. The data on hypoxia and tumorigenesis is incomplete, and confuses the main point the authors are making. In addition, the presentation of immunofluorescence images is poor, and makes it difficult to interpret some of the data. Finally, the appropriate controls are not routinely presented as part of the bar graphs, again making interpretation of the data very difficult.

- We appreciate again the referee's positive feedback and valuable suggestions for our studies. To improve our manuscript, we have replaced immunostaining data with 3D high quality images and included new quantification data by adequate comparative statistical analyses with appropriate controls. Moreover, we have included new data on hypoxia and rewritten that part in the revised manuscript (Fig 7, 8, EV4, EV5).

2nd Editorial Decision

13 November 2019

Thank you for submitting the revised version of your manuscript. It has now been seen by one of the original referees.

As you can see, the referee finds that the study is significantly improved during revision and recommends publication. Before I can accept the manuscript, I need you to address some minor points below:

- Please address the remaining minor concern of the referee by textual changes.

REFEREE REPORTS

Referee #1:

In this revised manuscript Soo Mi Ki and colleagues have done a very good job at addressing all my comments. I would like to congratulate the authors for their revised manuscript. The authors have greatly improved the quality of the experiments (with all needed controls), of the text and of the interpretation of the results. As such, I would highly recommend this work to be published in EMBO reports.

I only have one minor comment in the response to the question 7 to the reviewer 1: The authors have now included a picture of the mutant animal in the appendix figure S2I and refer to it as showing ciliary phenotypes. The curved body axis of this mutant is quite different from the typical cilia mutant in the sense that it is curled up and not down. It is also surprising that the curved body axis is different in the mutant versus the morpholino. This being said, all experiments performed in the context of this work are well controlled and this comment won't have a major impact on the conclusion of this manuscript. I would recommend the authors to be careful when describing the morphological phenotype of this mutant as a "typical ciliary phenotype" before further verifying cilia-related phenotype across tissues. And I agree that this is out of scope of this manuscript.

2nd Revision - authors' response

18 November 2019

We generally understand the referee #1's comments on the curvature direction of zebrafish body as ciliopathy phenotype, but disagree that only curled up is a typical ciliary phenotype in zebrafish. In fact, several previous studies have reported that zebrafish with ciliary defects show curled down of body axis [1-3], and even we have often observed both of our mutants and morphant zebrafish present curled down as well as curled up. As the referee #1 mentioned, this various phenotypes donot affect our current study and there is no related-sentences/words to change in the manuscript.

3rd Editorial Decision

29 November 2019

Thank you for submitting your revised manuscript. I have now looked at everything carefully and all looks fine. Therefore I am very pleased to accept your manuscript for publication in EMBO Reports.

Corresponding Author Name: Ji Eun Lee

Manuscript Number: EMBOR-2019-48290V1